# The Influence of Different Non-Conventional Yeasts on the Odour-Active Compounds of Produced Beers

**Paweł Satora ***🄳 **and Aneta Pater** 🄳

Department of Fermentation Technology and Microbiology, Faculty of Food Technology,
University of Agriculture, Balicka Street 122, 30-149 Kraków, Poland
* Correspondence: pawel.satora@urk.edu.pl

**Abstract:** The interest in new beer products, which has been growing for several years, forces technologists and brewers to look for innovative raw materials, such as hops, new sources of carbohydrates or yeast. The aim of the presented study was to evaluate the effect of selected *Saccharomyces* (*Saccharomyces paradoxus* (CBS 7302), *S. kudriavzevii* (CBS 3774), *S. cerevisiae* (Safbrew T-58)) and non-*Saccharomyces* yeast (*W. anomalus* (CBS 5759), *Ha. uvarum* (CBS 2768), *D. bruxellensis* (CBS 3429), *Z. bailii* (CBS 749), and *T. delbrueckii* (D10)) on the fermentation process, basic parameters and odour-active compounds of the produced beers. The chemical composition and key aroma components of the obtained beers were determined using various chromatographic methods (HPLC, GC-FID, GC-MS, and GC-O). We showed large differences between the key aroma components depending on the culture of microorganisms used. Forty different compounds that have an active impact on the creation of the aroma of beers were detected, among which the most important are: β-phenylethanol, ethyl hexanoate, ethyl 4-methylpentanoate, ethyl dihydrocinnamate and β-damascenone. We also found the presence of components specific to the yeast strain used, such as 2-methoxy-4-vinylphenol, γ-decalactone, methional, nerolidol and others. Among the analyzed yeasts, *S. kudriavzevii* and *W. anomalus* should be distinguished, which produced beers with intense fruity and floral aromas and were also characterized by favorable features for brewing. The *Z. bailii* strain also turned out to be interesting as a potential starter culture for the production of low-alcohol beers, significantly differing in sensory characteristics from the standard ones.

**Keywords:** non-conventional yeasts; beer; aroma compounds; olfactometry

## 1. Introduction

The beer market has changed significantly in recent years. Consumers stopped reaching for a beer just to quench their thirst. Beer lovers are looking for new flavors and clear but subtle accents. It has encouraged technologists and brewers to search for innovative technologies and materials. These include hops (aromatic, bitter, and universal), new sources of carbohydrates (buckwheat, amaranth, quinoa) and yeast (non-*Saccharomyces*) [1]. *Saccharomyces cerevisiae* and *S. pasteurianus* are the first recognized and widely used yeast for wort fermentation [2]. One of the main roles of these microorganisms during fermentation is the production of active aroma compounds in the form of metabolites, among which higher alcohols, esters and vicinal diketones are distinguished. These compounds have a direct impact on the quality of the finished beer [3]. However, the growing demand for alternative odour as well as low-alcohol beers has encouraged research into the potential benefits of non-conventional yeasts [4]. Non-*Saccharomyces* yeasts play a decisive role in the formation of aroma and taste compounds at the early stage of fermentation [5]. They give the beer a specific fruity and floral taste and different aroma profiles, so non-*Saccharomyces* yeast can be used to develop specialty beers [6]. All odour-active ingredients must fall within certain limits; otherwise, a single compound or group of compounds may dominate and upset the aroma balance [7]. Each type of beer has its own distinctive odour, which is

triggered by the selected strain of yeast. According to the literature, the *Dekkera bruxellensis* and *Hanseniaspora uvarum* yeast are characterized by the ability to produce many fruity esters. In addition, these microorganisms add aroma complexity and modify the mouth-feel [8]. *Torulaspora delbrueckii* yeast, on the other hand, produces during fermentation significant amounts of fruity aroma, characteristics for amyl alcohol [9]. Alternative fermentation yeasts, such as *Wickerhamomyces anomalus* [10], can be used to produce low-alcohol or alcohol-free beer. In the case of *Saccharomyces* yeast, this production involves either physically removing the alcohol or stopping fermentation. This process contributes to the reduction of the content of aroma compounds or their complete removal [11]. The use of *W. anomalus* yeast is therefore a good way to produce beers that still have the right taste and aroma [12].

Most scientists use methods based on quantitative analysis (GC-FID and GC-MS) to assess the impact of the microorganisms on the sensory characteristics of the obtained beer [13]. Due to the growing interest of consumers in beers with novel and specific aromas, it was decided to focus more on this aspect. For this purpose, in recent years, there have been more and more studies related to the sensory analysis of alcoholic beverages using gas chromatography with olfactometry (GC-O) [14]. Gas chromatography with olfactometry (GC-O) is a technique based on the sensory evaluation of the eluate from the chromatographic column to detect odour-active compounds [15]. This analysis uses the human sense of smell as a sensitive and selective detector of odour compounds separated by GC [16]. Carrying out the analysis is possible thanks to the presence of a snap-in, the so-called olfactometric port connected in parallel to conventional detectors, such as a flame ionization (FID) or electron ionization mass spectroscopy detector (EI-MS) [14].

The aim of the presented study was to evaluate the effect of selected *Saccharomyces* (*Saccharomyces paradoxus* (CBS 7302), *S. kudriavzevii* (CBS 3774), *S. cerevisiae* (Safbrew T-58)) and non-*Saccharomyces* yeast (*W. anomalus* (CBS 5759), *Ha. uvarum* (CBS 2768), *D. bruxellensis* (CBS 3429), *Z. bailii* (CBS 749), and *T. delbrueckii* (D10)) on the fermentation process, basic parameters and odour-active compounds of the produced beers. The chemical composition and key aroma components of the obtained beers were determined using various chromatographic methods (HPLC, GC-FID, GC-MS, and GC-O). Thanks to the obtained results, it was possible to determine how applied microorganisms could modify the intensity and character of the beer aroma.

## 2. Materials and Methods

### 2.1. Materials

*Saccharomyces paradoxus* (CBS 7302), *Saccharomyces kudriavzevii* (DSM 3774), *Wickehamomyces anomalus* (CBS 5759), *Hanseniaspora uvarum* (DSM 2768), *Dekkera bruxellensis* (DSM 3429), *Zygosaccharomyces bailii* (CBS 749), and *Torulospora delbrueckii* (D10) were used in experiments. The control culture was top-fermenting brewer's yeast of *Saccharomyces cerevisiae* Safbrew T-58.

### 2.2. Yeast Propagation

Yeast strains were cultured on slants with Sabouraud Dextrose LAB-AGAR medium in replicates at 28 °C for 24 h. Then, five loops of the slurry were inoculated into 10 mL of sterile Sabouraud Dextrose Broth. The flasks were secured with gauze and cotton wool plugs and left on a shaker for 24 h at 20 °C. In the next step, the slurry was poured into 40 mL of sterile Sabouraud Dextrose Broth and incubated on a shaker for 24 h at 20 °C.

After yeast propagation was completed, the number of yeast cells was evaluated using a Thoma chamber. On the basis of the obtained results, the amount of the individual yeast cultures was calculated to obtain $10^6$ cells per mL in inoculated hopped wort.

### 2.3. Hopped Wort Preparation and Fermentation

The malt extract (WES-Wolsztyn Poznań) was diluted with sterile distilled water at 20 °C to an extract of 12°P. Marynka variety hops (7.5% alpha acids) were added to boiling

wort, to receive 26 IBU. After 60 min of boiling, the wort was filtered using Whatman Grade 802 filter paper, and cooled to 20 °C. After inoculation, wort fermentation was carried out in 500 mL flasks containing 300 mL of wort. Three replicates for each variant were performed. The fermentation was carried out at 20 °C in anaerobic conditions (plugs with fermentation tubes filled with glycerine). The process was considered completed when daily mass losses of less than 0.20 g/L of the fermenting wort were obtained, and no significant changes in the apparent extract were obtained.

### 2.4. Fermentation Kinetics and Basic Parameters of the Obtained Beers

During fermentation, mass losses associated with the release of carbon dioxide (g/100 mL) were monitored. After its completion, yeast cells were separated from the beer by centrifugation (Centrifuge MPW-365, MPW Med. Instruments, Warsaw, Poland). The amount of yeast biomass in the sludge was determined using a moisture analyzer (Radwag MAC50, RADWAG, Radom, Poland). The content of ethanol in the samples was determined with an automatic beer analyzer (Alcolyzer, DMA 4500+, Anton Paar, Graz, Austria). Before starting the measurement, the samples were degassed and then filtered through a filter with diatomaceous earth.

### 2.5. Sugars and Organic Acid Analysis

The Shimadzu (Kyoto, Japan) NEXERA XR apparatus with the RF-20A refractometric detector was used for the analysis of sugars. The components were separated using an Asahipak NH2P-50 4.6 × 250 mm Shodex column (Showa Denko Europe, Munich, Germany) at 30 °C, 70% aqueous solution of acetonitrile was the mobile phase, and the isocratic elution program (0.8 mL/min) lasted 16 min. Standard curves for quantitative determinations were prepared with the appropriate standards: glucose, fructose, saccharose, maltose and glycerol (Sigma-Aldrich).

Analysis of organic acids was conducted on a Perkin-Elmer (USA) FLEXAR chromatograph with a UV-Vis detector. Before the analysis, beer samples were diluted five times with demineralized water and filtered through a syringe filter (0.45 μm). Lactic, acetic and succinic acids (Sigma-Aldrich) were analyzed using the Rezex ROA-Organic Acid Aminex HPX-87H column (300 mm, 18 cm × 7.8 mm). Samples were eluted isocratically at 40 °C with a mobile phase (0.005 M $H_2SO_4$) at a flow rate of 0.4 mL/min.

### 2.6. Main Volatile Compounds (HS-SPME-GC-FID) and Odour-Active Volatile Components (HS-SPME-GC-O)

In order to analyze the main volatile compounds, a 2 mL sample of wort/beer and an internal standard solution (0.57 mg/L 4-methyl-2-pentanol, 0.2 mg/L anethol and 1.48 mg/L of ethyl nonanoate (Sigma-Aldrich, Saint Louis, MO, USA) were placed into a 10 mL vial with 1 g of NaCl. The conditioned (250 °C for 1 h) SPME device (Supelco Inc., Bellefonte, PA, USA) coated with PDMS (100 μm) fiber was used for sampling. It was placed into the headspace under stirring (300 rpm) for 40 min at 40 °C. Next, the SPME device was desorbed in the injector port of the Hewlett Packard 5890 Series II chromatograph system for 3 min. A Rxi®-1 ms capillary column (Crossbond 100% dimethyl polysiloxane, 30 m × 0.53 mm × 0.5 μm) was used for separation of the analyzed volatiles. The column was heated using the following program: 35 °C for 4 min at an increment of 5 °C/min to 110 °C and then an increment of 20 °C/min to 230 °C, then maintaining a constant temperature for 4 min, and the detector temperature was 250 °C. Helium was the carrier gas at a constant flow of 1.0 mL/min.

Ethyl acetate, ethyl butanoate, ethyl 2-methylbutyrate, 3-methylbutyl acetate, ethyl 4-methylpentanoate, ethyl hexanoate, ethyl octanoate, 2-phenylethyl acetate, ethyl decanoate, acetaldehyde, 1-propanol, 2-methyl-1-propanol, 3-methyl-1-butanol, and 2-phenylethanol (Sigma-Aldrich) were used as standards for qualitative and quantitative evaluation, based on a comparison of their retention times and peak surface area reads with samples.

Odour-active volatiles of wort/beers were identified by olfactometry in the same GC system, using the same chromatographic conditions as mentioned above. Two mL sample of wort/beer with 1 g of NaCl placed in a 10 mL vial were exposed to a 2 cm divinylbenzene/carboxen/polydimethylsiloxane (DVB/CAR/PDMS) SPME fibre 50/30 μm (Supelco/SigmaAldrich, Bellafonte, PA, USA) for 40 min at 40 °C and were analysed using olfactory detection port (ODP-3, Gerstel, Linthicum Heights, MD, USA). Five trained analysts described the detected odours and their intensity using a 4-point scale (not detected, weak, moderate, and strong). For each wort/beer samples, the odour-active compounds were identified (based on the results of GC-MS analysis) and grouped into nine classes. We have distinguished fruity (FR), floral (FL), roasted (R), herbaceous (H), woody (W), vegetal (V), earthy (E), animal (A) and chemical (C) aroma group. The results of the aroma intensity of a given compound were calculated as the average intensities reported by the individual raters. Only those results with an average score of at least 0.6 points were presented.

### 2.7. Minor Volatile Compounds Using HS-SPME-GC-MS

Minor volatile analysis was performed as described by Januszek and Satora [17]. The wort/beer samples were prepared as mentioned in Section 2.6, with the addition of 1 g of NaCl and 0.1 mL of the internal standard. Three replicates per sample were prepared and analyzed.

An Agilent Technologies 7890B chromatograph system (Agilent Technologies, Santa Clara, CA, USA) interfaced with A Pegasus HT TOFMS (Time-of-Flight Mass Spectrometry) detector (LECO Corporation, St. Joseph, MI, USA) operated in electron ionization mode and equipped in MPS autosampler (Gerstel, Mülheim an der Ruhr, Germany) was used for the analyses. The volatiles were extracted and concentrated on a phase microextraction fiber coated with polydimethylsiloxane (100 μm PDMS, Supelco Inc., Bellefonte, PA, USA) in the sample headspace for 40 min at 40 °C. The compounds were desorbed in the injector port of chromatograph (250 °C, 3 min). Chromatographic separation conditions were same as described by Januszek and Satora [17]. Mass Spectra were recorded in the EI mode at an ionization voltage of 70 eV and a transfer line and ion source temperature of 250 °C.

Volatiles were identified using the National Institute of Standards and Technology (NIST) database and LRI (Linear Retention Indices), calculated based on a series of n-alkanes from C6 to C30. The quantitative identification of volatiles (shown in the Tables 2–4; Sigma-Aldrich) was based on the comparison of peak surface area of sample and standard chromatograms. Other detected components (marked with superscript, Tables 2–4) were determined semi-quantitatively by measuring the relative peak area of each identified compound, according to the NIST database, in relation to that of the chemically similar standard.

### 2.8. Sensory Analysis (QDA)

The sensory evaluation was performed by ten trained testers, employees of the Department of Fermentation Technology and Microbiology (five men and five women), aged 30 to 50 years. The aroma of the beers was characterized using the method of quantitative descriptive analysis (QDA), on a scale of 0–9 points. with an accuracy of 1 point. Awarding nine points. indicated that the evaluator sensed a highly intense aroma, while zero point indicated the lack of perception of a given aroma. The intensity of the following aromas was analyzed—fruity (red apple, pineapple, banana, and citrus), floral (rose, geranium, and honey), roasted (bready, yeasty, worty, and malty), herbaceous (beer, hoppy, spices, and herbs), woody (pine and resin), vegetal (boiled potatoes, boiled vegetables, onion, and garlic), earthy (earth, fungi, and mold), animal (cheesy, skunk, buttery, musk, and urine) and chemical (solvent, sulfuric, feces, and pharmacy).

*2.9. Statistical Analyses*

The presented results are the mean of 3-5 independent replicate experiments. The data were analysed using a one-way analysis of variance (ANOVA) and Duncan's post hoc test (SPSS Inc., Chicago, IL, USA).

## 3. Results and Discussion

*3.1. Fermentation Kinetics and Basic Parameters of the Obtained Beers*

Fermentation is the process when sugars available in the wort are converted by yeast into alcohol, carbon dioxide and heat [18]. The *S. cerevisiae* yeast strain is known especially for excellent fermentation properties, i.e., fast fermentation rate and high ethanol production [19]. In the present study, the *S. cerevisiae* strain was characterized by the shortest period of adaptation to the new environment and the highest rate of fermentation for the first 2 days of the process (Figure 1). In the case of non-*Saccharomyces* strains (*W. anomalus* and *D. bruxellensis*), the adaptation time was longer, and the highest amount of produced $CO_2$ was observed between the 2nd and 4th day of the process. These strains, apart from *S. cerevisiae*, fermented at a similar level from day 2 to the end of the process. This is also confirmed by the ethanol content results presented in Table 1. The ethanol content obtained by individual strains (*S. cerevisiae*, *W. anomalus*, and *D. bruxellensis*) did not differ statistically. The yeast *W. anomalus* shows a good ability to utilize maltose under both aerobic and anaerobic conditions [20]. Lee et al. [21] have proven that some wild strains of *W. anomalus* are able to make better use of maltose during the fermentation process than other commercial brewing yeast. This is also confirmed by the results presented in the article where this strain used over 90% of the maltose available in the wort for fermentation (Table 1). On the other hand, the *D. bruxellensis* strain is capable of producing acetic acid in the presence of oxygen. This feature is particularly desirable in the production of lambic beers, where it is increasingly used [22]. This was also confirmed in the presented article (Table 1), where *D. bruxellensis* produced the highest amount of acetic acid among all the analyzed strains. The best fermentation properties, and thus the use of the largest amount of maltose and the production of the largest amount of ethanol, were observed in the case of the *S. kudriavzevii* strain (Figure 1, Table 1). This strain produced the largest amount of $CO_2$ between the 6th and 8th day of the fermentation process. *S. kudriavzevii* is present in Belgian beers of spontaneous fermentation and is characterized by the rapid absorption of carbohydrates, high ethanol tolerance and the ability to ferment in the absence of oxygen [23]. Strains *H. uvarum* and *S. paradoxus* released an average of 2.4 g $CO_2$/100 mL during the fermentation process. *H. uvarum* yeast is characterized by very low fermentation power and efficiency, but it contributes to improving the complexity of the aroma of fermented beverages, such as beer or wine [23]. In the present study, this strain was characterized by a lower amount of carbon dioxide released during fermentation. However, it used approximately 98% of the available maltose and thus produced an ethanol content of 4.31 g/L. The lowest fermentation ability of the brewing wort was observed in the case of *Z. bailii* strain. Time of adaptation to the new environment for this strain lasted the longest, and after 8 days of fermentation, they produced approximately 80% less $CO_2$ than the *S. cerevisiae* and *S. kudriavzevii* strains. This strain was characterized by a low maltose fermentation capacity. During the entire process, *Z. bailii* strain used only 2% of the available maltose, producing a beer with a content of 1.73 g/L of ethanol. Taking into account the current interest in the production of low-alcohol beers due to the constantly developing market and lower tax burdens [24], the analyzed *Z. bailii* yeast strain shows potential for the production of low-alcohol beers.

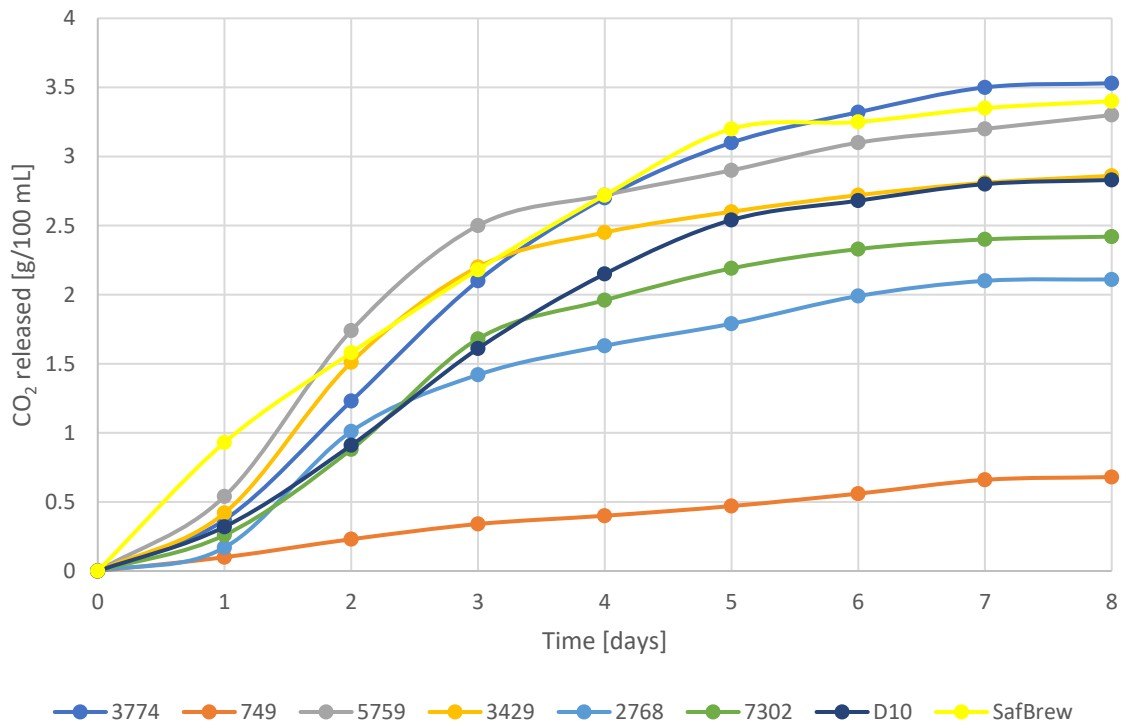

**Figure 1.** The kinetics of beer fermentation with different yeast strains; *Saccharomyces paradoxus* (7302), *Saccharomyces kudriavzevii* (3774), *Wickehamomyces anomalus* (5759), *Hanseniaspora uvarum* (2768), *Dekkera bruxellensis* (3429), *Zygosaccharomyces bailii* (749), *Torulospora delbrueckii* (D10), *Saccharomyces cerevisiae* Safbrew T-58 (Safbrew).

**Table 1.** The main parameters of beer produced by non-conventional yeasts.

| Parameters | Biomass [g/L] | Ethanol [% vol.] | Maltose [g/L] | Saccharose [g/L] | Fructose [g/L] | Glucose [g/L] | Glycerol [g/L] | Lactic Acid [mg/L] | Acetic Acid [mg/L] | Succinic Acid [mg/L] |
|---|---|---|---|---|---|---|---|---|---|---|
| Wort | - | - | 82.8 d (±2.7) | 10.5 e (±1.2) | 3.0 b (±1.3) | 8.7 e (±1.5) | 0.0 a (±0.0) | 0.0 a (±0.0) | 0.0 a (±0.0) | 0.0 a (±0.0) |
| *Saccharomyces cerevisiae* Safbrew T-58 | 1.23 e (±0.12) | 4.72 c (±0.02) | 4.9 b (±0.4) | 0.0 a (±0.0) | 5.5 c (±0.1) | 2.9 c (±0.1) | 8.4 de (±0.4) | 118.0 def (±6.0) | 264.0 cd (±26.0) | 0.0 a (±0.0) |
| *Saccharomyces paradoxus* (CBS 7302) | 1.25 e (±0.04) | 3.51 b (±1.65) | 8.3 bc (±1.1) | 3.2 c (±0.1) | 0.a (±0.0) | 2.2 bc (±0.1) | 8.6 de (±0.9) | 110.0 cde (±22.0) | 167.3 b (±30.5) | 72.3 cd (±26.5) |
| *Saccharomyces kudriavzevii* (CBS 3774) | 0.93 b (±0.01) | 5.97 d (±0.07) | 1.1 a (±0.1) | 0.0 a (±0.0) | 0.0 a (±0.0) | 0.2 a (±0.2) | 4.3 b (±0.3) | 78.3 b (±6.5) | 227.0 c (±21.0) | 16.0 ab (±27.7) |
| *Wickehamomyces anomalus* (CBS 5759) | 0.76 a (±0.01) | 4.26 bc (±0.04) | 5.5 bc (±0.2) | 1.7 b (±0.9) | 0.8 a (±0.7) | 10.0 f (±0.4) | 8.4 de (±1.0) | 93.0 bc (±6.0) | 344.0 f (±10.0) | 106.0 d (±29.0) |
| *Dekkera bruxellensis* (CBS 3429) | 1.03 c (±0.01) | 4.60 bc (±0.03) | 1.2 a (±0.1) | 0.0 a (±0.0) | 0.0 a (±0.0) | 1.2 b (±0.0) | 8.8 de (±0.8) | 140.3 f (±5.5) | 1356.3 g (±58.5) | 30.0 ab (±11.0) |
| *Hanseniaspora uvarum* (CBS 2768) | 1.21 e (±0.00) | 4.31 bc (±0.03) | 1.2 a (±0.1) | 0.0 a (±0.0) | 0.0 a (±0.0) | 1.4 b (±0.0) | 7.8 d (±0.5) | 127.3 ef (±8.5) | 296.0 de (±7.0) | 53.3 bc (±20.5) |
| *Zygosaccharomyces bailii* (CBS 749) | 0.91 b (±0.01) | 1.73 a (±0.03) | 81.8 d (±4.1) | 4.5 d (±0.7) | 0.0 a (±0.0) | 6.2 d (±0.1) | 6.6 c (±0.3) | 134.3 f (±6.5) | 316.3 ef (±9.5) | 48.0 bc (±8.0) |
| *Torulospora delbrueckii* (CBS D10) | 1.12 d (±0.03) | 4.36 bc (±0.06) | 1.3 a (±0.1) | 4.1 cd (±0.7) | 5.1 c (±0.3) | 6.5 d (±0.2) | 9.3 e (±0.5) | 101.0 cd (±24.5) | 224.0 c (±4.0) | 221.3 e (±22.5) |
| [1] Sig | *** | *** | *** | *** | *** | *** | *** | *** | *** | *** |

[1] Significance; *** indicates significance at a level of <0.005, respectively, by the least significant difference. Values with different letters (a–g) in the same column indicate statistical differences, according to the Duncan test ($p < 0.05$).

### 3.2. Key Aroma Compounds of Hopped Wort

Malt extract was used for wort preparation. It is used, especially in smaller breweries as an additive source of extract, because it is much cheaper than malt and gives the product beneficial properties [25]. According to our previous investigations, the wort prepared from the extract is characterized by a deeper color and a specific, positive aroma. There are also

no turbidities and sediments, characteristic of the wort made from malt. The preparation of the wort from the malt extract significantly influenced its aroma.

Roasted aromas (including such aroma notes as malty, worty, caramellic, bready and yeasty) accounted for almost half of all aromas detected during the olfactometric analysis of the hopped wort used for fermentation (Table 2, Figure 2). The Maillard Reaction Products (MRPs) are essential contributors to malt flavour and colour [26]. The literature shows that some Strecker aldehydes, particularly 2-methylbutanal, 3-methylbutanal, benzeneacetaldehyde and methional, have an important role in the attributes associated with flavor of wort. These aldehydes have exceptionally low odor thresholds and impart potent worty, malty, and cocoa-like aromas even at low concentrations [27]. This was indeed confirmed by our research, i.e., four components—Strecker aldehydes (methional, 3-methylbutanal and benzeneacetaldehyde) and other MRPs (such as maltol) were in the top five most intense aromas during olfactometric analyses of the prepared hopped wort (Figure 2). Such an intense aroma of methional may be a characteristic feature of worts obtained from malt extract. Ditrych et al. [28] showed that malt is the main source of aging aldehydes, such as methional, introduced into the wort during the boiling process and its level increases during storage. Therefore, it should be assumed that malt extract, usually stored long before use, may be a potentially important source of this type of substance. Methional is characterized by the aroma of boiled potatoes and in higher concentrations is negatively correlated with the aroma quality of beer [29]. Despite the high intensity of the aroma in the olfactometric analysis, this compound did not negatively affect the aroma of the wort (Figure 3) or produced beers (Figure 4). The second most intense aroma in the hopped wort was 3-methylbutanal (3-MB). Together with 2-methylbutanal (2-MB), it is an important aging compound and wort off-flavor with a typical sweet, bread-like aroma [30]. Benzeneacetaldehyde (sweet and cocoa), maltol (sweet and caramellic aroma) and 5,5-dimethyl-2(5H)-furanone were also significant for the aroma of the wort, resulting from the Strecker degradation. The key aromas of hopped wort also included vanillin with a sweet, vanilla-cream aroma.

**Table 2.** The key aroma components of hopped wort used for the fermentation.

| Compound | LRI [1] | Ion [m/z] | Threshold [2] [μg/L] | Concentration [μg/L] | OAV [3] | Aroma descriptor [4] |
|---|---|---|---|---|---|---|
| 3-Methylbutanal | 634 | 44 | 0.2 | 9.7 | 48.3 | Bready, fruity [R] |
| 2-Methylbutanal [x] | 650 | 44 | 12.5 | 5.0 | 0.4 | |
| 2,3-Pentanedione | 675 | 43 | 900 | 1.6 | 0.0 | Sweet, cheesy, bready [A] |
| Dimethyl disulfide | 734 | 94 | 0.16 | 1.4 | 8.8 | Sulfurous, earthy, mushroom [E] |
| Furfural | 815 | 96 | 250 | 2.5 | 0.0 | |
| 3-Methylbutanoic acid | 831 | 60 | 22 | 2.9 | 0.1 | Earthy, mushroom, cheesy [E] |
| Methional | 894 | 48 | 0.2 | 6.5 | 32.5 | Boiled potatoes [V] |
| 5,5-Dimethyl-2(5H)-furanone [x] | 908 | 97 | | 3.4 | | Worty [R] |
| Benzaldehyde | 926 | 77 | 350 | 1.2 | 0.0 | |
| Dimethyl trisulfide | 945 | 126 | 0.005 | 0.9 | 178.8 | Sulfurous, cooked onion [V] |
| Hexanoic acid | 963 | 60 | 3000 | 2.9 | 0.0 | |
| Octanal | 979 | 43 | 0.7 | 3.6 | 5.2 | Aldehydic, solvent [C] |
| β-Myrcene | 1000 | 93 | 13 | 0.8 | 0.1 | |
| Benzeneacetaldehyde | 1029 | 91 | 4 | 23.9 | 6.0 | Green, sweet, cocoa [FR] |
| 2-Ethyl-1-hexanol [x] | 1045 | 57 | 270000 | 3.8 | 0.0 | |
| Acetophenone | 1042 | 105 | 65 | 4.5 | 0.1 | |
| 1-Octanol | 1068 | 56 | 110 | 0.8 | 0.0 | |

**Table 2.** *Cont.*

| Compound | LRI [1] | Ion [m/z] | Threshold [2] [µg/L] | Concentration [µg/L] | OAV [3] | Aroma descriptor [4] |
|---|---|---|---|---|---|---|
| cis-Linaloloxide | 1078 | 59 | 7 | 0.9 | 0.1 | |
| Maltol | 1080 | 126 | 35000 | 8.6 | 0.0 | Sweet, caramellic, bready [R] |
| Nonanal | 1085 | 57 | 1 | 2.8 | 2.8 | |
| Linalool | 1087 | 71 | 6 | 1.4 | 0.2 | Floral, earl grey, sweet [FL] |
| 1-Nonanol | 1156 | 56 | 50 | 1.0 | 0.0 | |
| Octanoic acid | 1153 | 60 | 3000 | 3.1 | 0.0 | |
| α-Terpineol | 1179 | 93 | 330 | 0.4 | 0.0 | |
| Decanal | 1185 | 82 | 0.1 | 2.0 | 20.0 | Sweet, aldehydic, floral [FL] |
| Geraniol | 1243 | 69 | 4 | 4.5 | 1.1 | Sweet, floral [FL] |
| Decanol | 1252 | 55 | 400 | 3.2 | 0.0 | |
| Methyl geranate | 1305 | 69 | 21 | 27.2 | 1.3 | Hoppy, sweet, floral [H] |
| γ-Nonalactone [x] | 1323 | 85 | 30 | 3.1 | 0.1 | |
| n-Decanoic acid | 1338 | 60 | 10000 | 0.9 | 0.0 | |
| Vanillin [x] | 1371 | 151 | 20 | 2.1 | 0.1 | Sweet, vanilla, creamy [H] |
| Damascenone | 1373 | 121 | 0.002 | 3.9 | 1949 | Fruity, plum [FR] |
| Dodecanal | 1392 | 82 | 2 | 1.0 | 0.5 | |
| γ-Decalactone | 1424 | 85 | 11 | 3.1 | 0.3 | |
| Caryophyllene | 1437 | 93 | 64 | 9.8 | 0.2 | |
| 1-Dodecanol | 1463 | 55 | 1000 | 14.9 | 0.0 | |
| Humulene | 1484 | 93 | 50 | 4.0 | 0.0 | |
| α-Farnesene | 1503 | 93 | 87 | 21.0 | 0.2 | Hoppy, citrus, floral [H] |

[1] LRI—linear retention index; the amount of components was determined. [2] Threshold in beer [31]. [3] OAV–Odor; Activity Values have been identified by color; OAV > 1 . [4] Aroma descriptor perceived at the sniffing port of the GC-O. [x]—determined semi-quantitatively by measuring the relative peak area of each identified compound, according to the NIST database, in relation to that of the chemically similar standard. The aroma group of detected aroma descriptors was signed by letters in brackets—fruity (FR), floral (FL), roasted (R), herbaceous (H), woody (W), vegetal (V), earthy (E), animal (A) and chemical (C).

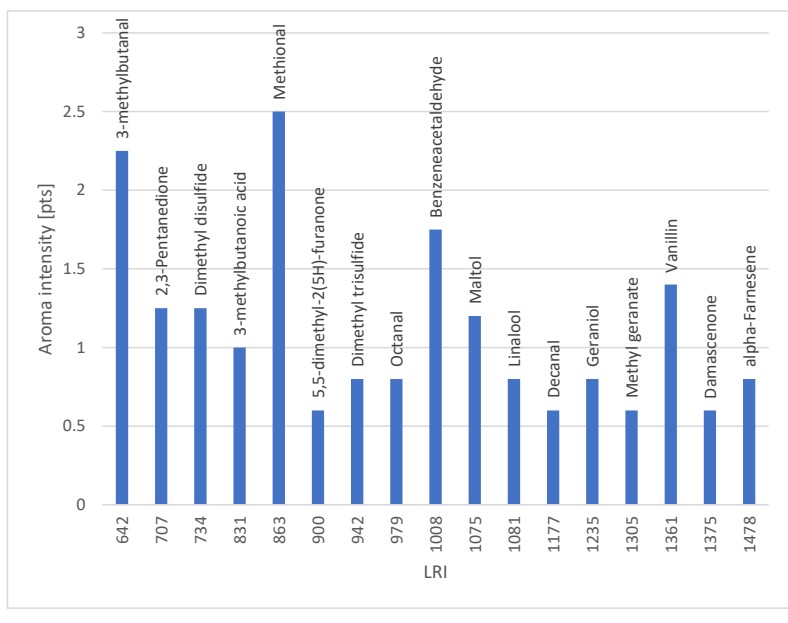

**Figure 2.** Odour-active compounds and their intensities in hopped unfermented wort.

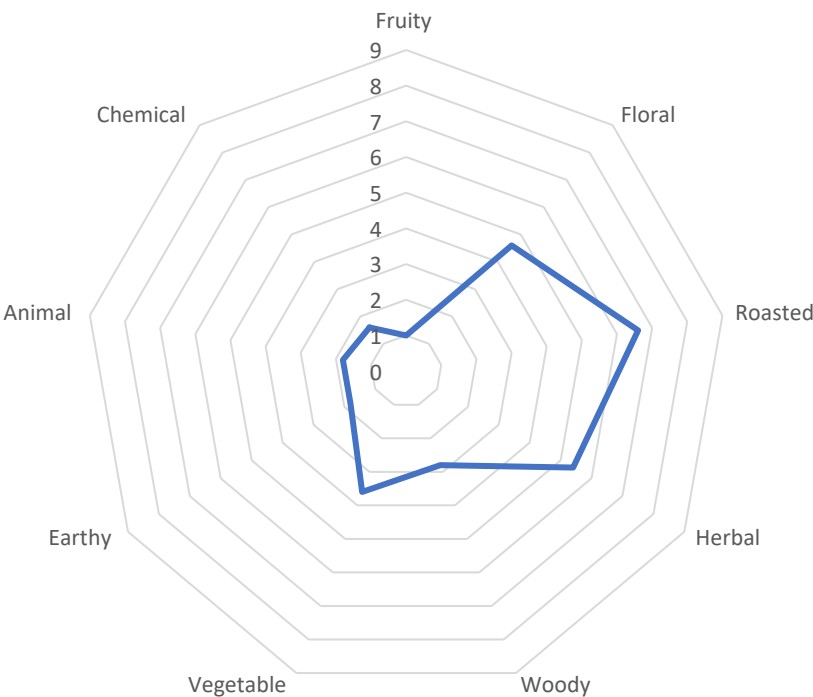

**Figure 3.** Sensory analysis (QDA) of hopped unfermented wort.

Compounds derived from hops had a relatively small share in the aroma of the hopped wort. This was due to the relatively low degree of hopping of the wort (26 IBU), so that unconventional yeast cultures more sensitive to hop-derived components would not have a problem with growth on this type of medium. Five compounds derived from hops-linalool, geraniol, methyl geranoate, damascenone and alpha-farnesene were detected during the olfactometric analysis of the hopped wort [Figure 2]. Most of them were characterized by a floral, citrus, hoppy aroma, while damascenone was distinguished by the aroma of sweet plum. Quantitatively, hopped wort contained the largest amounts of methyl geranate (27.2 μg/L) and alpha-farnesene (21.0 μg/L) [Table 2]. Considering the high content of the former compound, the hops used should be classified as methyl geranate-rich hops [32].

The sensory analysis of the aroma by QDA method confirmed the GC-O results (Figure 3). The highest scores (6.6 pts) on a 9-point scale were obtained by roasted notes, followed by herbal (5.4) and floral notes (4.6). Vegetable notes associated with the presence above the detection threshold of sulfur components, such as methional, dimethyl disulfide and dimethyl trisulfide, scored lower. At the same time, they did not have a negative impact on the overall assessment of the aroma of the hopped wort.

*3.3. Key Aroma Compounds of Beers Produced with Non-Conventional Yeasts*

As a result of the conducted olfactometric analysis, 40 odour-active components were detected in the obtained beers (Tables 3–5). This was almost twice as many as the number of compounds (29) for which the calculated OAV exceeded one. The aroma notes have changed significantly from those detected in the hopped, unfermented wort. In most cases, fruit and flower aromas were the most intense.

**Table 3.** Alcohols and esters in beers produced with different non-conventional yeasts.

| [µg/L] | m/z | LRI [2] | Threshold [3] | Saccharomyces cerevisiae Safbrew T-58 | Saccharomyces paradoxus CBS 7302 | Saccharomyces kudriavzevii CBS 3774 | Wickerhamomyces anomalus CBS 5759 | Dekkera bruxellensis CBS 3429 | Hanseniaspora uvarum CBS 2768 | Zygosaccharomyces bailii CBS 749 | Torulospora delbrueckii D10 | Sig [1] | GC-O Descriptors [4] | Detector |
|---|---|---|---|---|---|---|---|---|---|---|---|---|---|---|
| | | | | | | | Alcohols | | | | | | | |
| 1-Propanol | 31 | 546 | 9000 | 8864 a | 9042 a | 12175 b | 9187 a | 10600 a | 10108 ab | 7028 a | 9366 a | * | | FID MS |
| 2-Methyl-1-Propanol | 43 | 607 | 8300 | 7428 b | 6895 b | 6557 a | 10441 b | 6646 a | 5894 a | 5350 a | 5425 a | * | bready, floral, solvent [R] | FID MS |
| 3-Methyl-1-Butanol | 55 | 722 | 1000 | 1520 a | 1199 a | 1110 b | 1382 b | 1350 d | 1401 b | 894 c | 1399 b | *** | bready, alcoholic, fruity [R] | FID MS |
| 2-Methyl-1-Butanol | 56 | 725 | 15.9 | 136 cd | 98 bc | 166 d | 133 cd | 162 d | 134 cd | 58 a | 74 ab | *** | | MS |
| 4-Methyl-1-Pentanol ˣ | 56 | 819 | 820 | 0.4 bc | 0.3 a | 0.5 cd | 0.6 d | 0.4 bc | 0.5 cd | 0.8 e | 0.3 ab | *** | sweet fruity nutty [FR] | MS |
| 3-Methyl-1-Hexanol ˣ | 43 | 919 | | 1.1 bc | 2.2 d | 1.2 bc | 5.5 e | 1.6 c | 1.0 b | 0.0 a | 0.0 a | *** | sweet fruity solvent [FR] | MS |
| 2-Ethyl-1-Hexanol ˣ | 57 | 1010 | 270000 | 28 a | 14 a | 58 ab | 80 b | 16 a | 18 a | 28 a | 42 ab | * | | MS |
| 1-Octanol | 56 | 1057 | 110 | 12.9 bcd | 4.2 ab | 15.1 cd | 17.2 d | 9.3 b | 10.5 bc | 11.6 bc | 9.9 b | *** | herbal solvent wood [H] | MS |
| β-Phenylethanol | 91 | 1070 | 1000 | 7825 b | 8978 b | 9981 b | 10934 b | 14088 a | 15600 a | 2208 c | 1860 c | *** | rose petals {FL] | FID MS |
| 1-Nonanol | 56 | 1155 | 50 | 0.0 a | 0.0 a | 3.6 c | 2.2 b | 0.0 a | 0.3 a | 4.0 c | 0.0 a | *** | floral rose clean [FL] | MS |
| Decanol | 70 | 1255 | 400 | 11 b | 15 b | 14 b | 12 b | 14 b | 17 b | 2 a | 2 a | *** | fatty waxy floral [FL] | MS |
| 2-Methoxy-4-Vinylphenol | 135 | 1291 | 3 | 31.4 bc | 29.9 bc | 22.2 b | 35.5 c | 109.8 d | 3.5 a | 26.9 bc | 34.3 c | *** | spicy smoky woody [H] | MS |
| 1-Dodecanol | 55 | 1460 | 1000 | 27 | 22 | 30 | 38 | 25 | 20 | 39 | 10 | ns | | MS |
| 1-Tetradecanol | 43 | 1665 | 5000 | 1.1 bcd | 0.7 ab | 1.6 cd | 1.7 d | 1.4 bcd | 1.2 bcd | 0.9 abc | | * | | MS |
| | | | | | | | Esters | | | | | | | |
| Ethyl Acetate | 43 | 597 | 5000 | 9188 a | 10444 a | 13427 a | 2304 a | 3471 a | 17070 b | 4151 a | 15012 a | * | floral, solvent [FL] | FID MS |
| Ethyl Propanoate | 29 | 691 | 10 | 1.0 ab | 4.1 c | 2.2 ab | 0.8 a | 1.7 ab | 1.6 ab | 2.7 bc | 1.6 ab | * | sweet, fruity, pineapple [FR] | MS |
| 2-Methylpropyl Acetate | 43 | 755 | 66 | 1.3 ab | 1.1 ab | 1.7 bc | 0.6 ab | 2.7 c | 0.6 ab | 0.0 a | 1.5 bc | ** | sweet fruity banana [FR] | MS |
| Ethyl Butanoate | 71 | 784 | 1 | 21.6 a | 31.7 a | 51.5 b | 47.6 b | 21.5 a | 42.8 b | 24.8 a | 22.9 a | ** | fruity pineapple [FR] | FID MS |
| Ethyl Lactate | 45 | 791 | 14000 | 0.5 a | 0.0 a | 5.6 b | 0.6 a | 0.0 a | 0.2 a | 0.6 a | 0.0 a | * | sweet fruity malty [FR] | MS |
| Ethyl 2-Methylbutyrate | 57 | 836 | 0.3 | 235 a | 72 b | 67 b | 317 c | 86 b | 104 b | 201 a | 287 c | *** | fruity apple red fruits [FR] | FID MS |
| 3-Methylbutyl Acetate | 43 | 861 | 13 | 121 b | 147 b | 216 a | 124 b | 223 a | 122 b | 16 c | 382 d | *** | fruity solvent honey [FR] | FID MS |
| Ethyl Pentanoate | 29 | 882 | 1.5 | 34.4 c | 1.2 a | 5.3 b | 2.2 a | 1.8 a | 4.7 b | 2.2 a | 4.6 b | *** | sweet fruity apple [FR] | MS |

**Table 3.** *Cont.*

| [µg/L] | m/z | LRI [2] | Threshold [3] | *Saccharomyces cerevisiae* Safbrew T-58 | *Saccharomyces paradoxus* CBS 7302 | *Saccharomyces kudriavzevii* CBS 3774 | *Wickehamomyces anomalus* CBS 5759 | *Dekkera bruxellensis* CBS 3429 | *Hanseniaspora uvarum* CBS 2768 | *Zygosaccharomyces bailii* CBS 749 | *Torulospora delbrueckii* D10 | Sig [1] | GC-O Descriptors [4] | Detector |
|---|---|---|---|---|---|---|---|---|---|---|---|---|---|---|
| Ethyl 2,4-Hexadienoate ˣ | 95 | 899 | | 2.7 b | 6.6 d | 6.7 d | 2.4 b | 4.6 c | 10.3 e | 0.0 a | 2.1 b | *** | sweet fruity ethereal [FR] | MS |
| Ethyl 4-Methylpentanoate | 88 | 946 | 0.75 | 7.7 a | 10.2 bc | 9.3 b | 13.2 d | 12.7 d | 17.8 e | 10.7 c | 8.8 ab | *** | red apple red fruits [FR] | FID MS |
| Ethyl Hexanoate | 88 | 980 | 1 | 7.2 b | 11.1 c | 18.0 a | 8.3 bc | 10.0 bc | 21.6 d | 3.6 e | 14.7 f | *** | red apple [FR] | FID MS |
| Hexyl Acetate | 43 | 989 | 2 | 0.1 ab | 0.1 ab | 0.0 a | 0.0 a | 0.2 b | 0.0 a | 0.0 a | 0.5 c | *** | | MS |
| 2-Phenylethyl Formate ˣ | 104 | 1112 | | 0.0 a | 0.0 a | 1.8 b | 3.6 c | 0.0 a | 3.6 c | 2.0 c | 0.0 a | *** | rose herbal [FL] | MS |
| Ethyl Octanoate | 88 | 1179 | 70 | 31.6 b | 32.3 b | 53.5 c | 70.5 a | 68.6 a | 10.2 d | 12.3 d | 34.5 b | *** | sweet fruity winey [FR] | FID MS |
| β-Phenylethyl Acetate | 104 | 1218 | 250 | 171 a | 111 a | 322 b | 221 a | 120 a | 118 a | 2 c | 614 d | *** | rose honey sweet [FL] | FID MS |
| Ethyl Dihydrocinnamate | 104 | 1324 | 0.07 | 1.50 | 2.39 | 2.87 | 3.44 | 4.00 | 4.62 | 3.04 | 3.62 | *** | plum honey rummy {FR} | MS |
| Ethyl Decanoate | 88 | 1380 | 200 | 2816 a | 2251 a | 2782 a | 2659 a | 266 b | 2705 a | 12 c | 4937 d | ** | sweet fruity brandy {FR} | FID MS |
| 3-Methylbutyl Octanoate | 70 | 1438 | 2000 | 7.74 | 7.4 c | 14.6 d | 7.6 c | 1.7 ab | 8.6 c | 0.1 a | 5.0 bc | *** | | MS |
| 2-Methylbutyl Octanoate ˣ | 127 | 1444 | | 1.4 b | 1.4 b | 2.0 b | 1.1 ab | 1.0 ab | 1.4 b | 0.2 a | 3.3 c | *** | | MS |
| Propyl Decanoate ˣ | 61 | 1497 | | 1.1 de | 0.5 b | 1.2 de | 1.4 e | 0.0 a | 0.9 cd | 0.0 a | 0.6 bc | *** | | MS |
| Ethyl Dodecanoate | 88 | 1576 | 1500 | 133 c | 33 ab | 150 c | 84 bc | 15 a | 117 c | 8 a | 259 d | *** | | MS |
| 3-Methylbutyl Decanoate | 70 | 1650 | 3000 | 14 c | 8 b | 18 c | 16 c | 2 a | 16 c | 0 a | 7 b | *** | | MS |
| 2-Methylbutyl Decanoate ˣ | 119 | 1655 | | 1.2 c | 1.6 c | 1.2 c | 1.3 c | 0.1 ab | 1.3 c | 0.0 a | 0.6 b | *** | | MS |

[1] Significance; *, **, *** indicates significance at a level of 0.05–0.01, 0.01–0.005 and <0.005, respectively, by the least significant difference. Values with different superscript roman letters (a–f) in the same raw indicate statistical differences, according to the Duncan test ($p < 0.05$). [2] LRI—linear retention index; the amount of components was determined. [3] Threshold in beer [31]. OAV > 1 . [4] Aroma descriptor perceived at the sniffing port of the GC-O. ˣ—determined semi-quantitatively by measuring the relative peak area of each identified compound, according to the NIST database, in relation to that of the chemically similar standard. Aroma group of detected aroma descriptors was signed by letters in brackets—fruity (FR), floral (FL), roasted (R), herbaceous (H), woody (W), vegetal (V), earthy (E), animal (A) and chemical (C). SD < 5% Fruity and floral notes dominated in the analyzed beers, they characterized 17 and 13 of the odor-active compounds detected, respectively. These were also the notes of the highest intensity. In addition to the 2-phenylethanol discussed above, different esters, carbonyl compounds and terpenes were also characterized by such aromas [Tables 3–5].

**Table 4.** Odour-active compounds and heatmap of their intensities detected by GC-O in beers produced with different yeast.

| Compound | LRI [1] | *Saccharomyces cerevisiae* Safbrew T-58 | *Saccharomyces paradoxus* CBS 7302 | *Saccharomyces kudriavzevii* CBS 3774 | *Wickehamomyces anomalus* CBS 5759 | *Dekkera bruxellensis* CBS 3429 | *Hanseniaspora uvarum* CBS 2768 | *Zygosaccharomyces bailii* CBS 749 | *Torulospora delbrueckii* D10 |
|---|---|---|---|---|---|---|---|---|---|
| Ethyl Acetate | 598 | 0 | 0 | 0 | 0 | 0 | 1 | 0.6 | 0.8 |
| 2-methyl-1-propanol | 608 | 0.6 | 0.6 | 0 | 0.8 | 0 | 0 | 0.8 | 0 |
| Ethyl propanoate | 691 | 0 | 1 | 0.6 | 0 | 0.6 | 0 | 0.8 | 0 |
| 3-methyl-1-butanol | 723 | 1.4 | 1 | 1 | 0.6 | 0.8 | 1.4 | 1.8 | 1.4 |
| Dimethyl disulfide | 743 | 0 | 0.8 | 0 | 1 | 1.2 | 0 | 0 | 0 |
| Isobutyl acetate | 755 | 1.2 | 0.6 | 0.6 | 2 | 1 | 1 | 0 | 0.6 |
| Ethyl butanoate | 784 | 0.6 | 0.8 | 1.2 | 1.2 | 0 | 1 | 0.8 | 0.6 |
| Ethyl lactate | 794 | 0.6 | 0 | 0 | 1 | 0.6 | 0 | 0 | 0 |
| 4-methyl-1-pentanol | 819 | 0.6 | 0 | 1.2 | 0.6 | 1.2 | 0 | 0 | 0 |
| Ethyl 2-methylbutyrate | 835 | 2 | 0.6 | 0 | 2.2 | 0.8 | 1.2 | 2 | 1.6 |
| 3-Methylbutyl acetate | 856 | 0 | 0 | 0 | 0 | 0.6 | 0 | 0 | 0.6 |
| Methional | 866 | 0 | 0.8 | 0.6 | 0.8 | 1.8 | 0.8 | 2.6 | 0.6 |
| Ethyl pentanoate | 882 | 1.6 | 0 | 0.6 | 0 | 0 | 0.6 | 0 | 0.6 |
| Ethyl 2,4-hexadienoate | 900 | 0 | 0.6 | 0.6 | 0 | 0 | 0.8 | 0 | 0 |
| 3-Methyl-1-hexanol | 918 | 0 | 0.8 | 0 | 1.4 | 0 | 0 | 0 | 0 |
| Dimethyl trisulfide | 933 | 0 | 0 | 0.8 | 1 | 0.8 | 0.6 | 0 | 0 |
| Ethyl 4-methylpentanoate | 944 | 0.6 | 1.2 | 1 | 1.8 | 1.8 | 2.4 | 1.8 | 1 |
| Ethyl hexanoate | 976 | 1.8 | 1.8 | 2.4 | 1.2 | 2 | 2.4 | 1.2 | 2 |
| Benzeneacetaldehyde | 1005 | 0.6 | 0 | 0 | 1 | 0.6 | 0.6 | 1 | 0 |
| Acetophenone | 1040 | 0 | 0 | 0.8 | 0 | 0 | 0.6 | 1.4 | 0 |
| 1-Octanol | 1060 | 0.8 | 0 | 1 | 0.8 | 0 | 0.6 | 0.6 | 0.6 |
| Nonanal | 1072 | 0.6 | 1.4 | 0.6 | 1 | 0.6 | 0.6 | 0.6 | 1.2 |
| Phenylethyl Alcohol | 1085 | 1.6 | 2 | 2.4 | 2.6 | 2.6 | 2.8 | 2.2 | 1.6 |
| cis-Rose oxide | 1095 | 0.8 | 0 | 0.6 | 1 | 1.6 | 0 | 1.2 | 0.6 |
| 2-Phenylethyl formate | 1112 | 0 | 0 | 0.8 | 1.2 | 1.2 | 0 | 0.6 | 0 |
| 1-Nonanol | 1159 | 0 | 0 | 0.8 | 0 | 0 | 0 | 1.2 | 0 |
| Ethyl octanoate | 1174 | 1 | 1 | 1 | 1.8 | 1.2 | 0.6 | 1.2 | 1 |
| Decanal | 1183 | 1 | 1 | 0.8 | 0.6 | 0.8 | 0.8 | 0 | 1 |
| Citronellol | 1207 | 0 | 0 | 0 | 0 | 0 | 0 | 0.8 | 0 |
| 2-Phenylethyl acetate | 1218 | 0 | 0 | 1 | 0 | 0 | 0 | 0 | 1.2 |

**Table 4.** *Cont.*

| Compound | LRI [1] | *Saccharomyces cerevisiae* Safbrew T-58 | *Saccharomyces paradoxus* CBS 7302 | *Saccharomyces kudriavzevii* CBS 3774 | *Wickehamomyces anomalus* CBS 5759 | *Dekkera bruxellensis* CBS 3429 | *Hanseniaspora uvarum* CBS 2768 | *Zygosaccharomyces bailii* CBS 749 | *Torulospora delbrueckii* D10 |
|---|---|---|---|---|---|---|---|---|---|
| Geraniol | 1235 | 0 | 0 | 1 | 1.2 | 0.6 | 0 | 0 | 0 |
| Decanol | 1255 | 0 | 1.6 | 0.6 | 0 | 0 | 1.4 | 0 | 0 |
| 2-Methoxy-4-vinylphenol | 1291 | 0.6 | 0 | 0 | 0.8 | 1.2 | 0 | 0 | 0.6 |
| Ethyl dihydrocinnamate | 1324 | 1.4 | 1.4 | 1.6 | 2 | 2.2 | 2.4 | 1.8 | 2 |
| Damascenone | 1373 | 0.6 | 1.6 | 1.4 | 1.8 | 1.8 | 0.6 | 1.6 | 0.6 |
| Ethyl decanoate | 1380 | 1 | 1.2 | 1.2 | 1.2 | 0.6 | 1 | 0 | 1.4 |
| Dodecanal | 1392 | 0.8 | 1 | 1 | 0.6 | 1.2 | 0.8 | 1 | 1 |
| γ-Decalactone | 1443 | 0.6 | 0 | 0.6 | 0 | 0 | 0 | 0 | 0.6 |
| β-Ionone | 1483 | 0 | 0 | 0.8 | 0 | 0 | 0 | 0 | 0 |
| Nerolidol | 1562 | 0.8 | 0.6 | 0 | 1 | 0 | 0 | 0 | 0 |

[1] LRI—linear retention index. The lowest intensity of aroma in a column is in the darkest red and the highest is in the darkest green. SD < 5%.

| min | | | | max |
|---|---|---|---|---|

**Table 5.** Terpenes and other volatiles in beers produced with different non-conventional yeasts.

| [µg/L] | m/z | LRI [2] | Threshold [3] | *S. cerevisiae* Safbrew T-58 | *S. paradoxus* CBS7302 | *S. kudriavzevii* CBS3774 | *Wickerhammomyces anomalus* CBS5759 | *Hanseniaspora uvarum* CBS2768 | *Dekkera bruxellensis* CBS3429 | *Zygosaccharomyces bailii* CBS729 | *Torulaspora delbrueckii* D10 | Sig. [1] | GC-O descriptors [4] |
|---|---|---|---|---|---|---|---|---|---|---|---|---|---|
| | | | | | | | Terpenes | | | | | | |
| Cis-Linaloloxide | 59 | 1062 | 7 | 0.00 a | 0.08 c | 0.04 b | 0.20 d | 0.09 c | 0.06 bc | 0.08 c | 0.00 a | *** | |
| Linalool | 71 | 1087 | 6 | 0.07 a | 0.05 a | 0.44 c | 0.35 bc | 0.27 b | 0.00 a | 0.37 bc | 0.00 a | *** | |
| Cis Rose Oxide | 139 | 1095 | 0.5 | 0.73 b | 0.00 a | 0.66 b | 0.70 b | 0.94 c | 0.00 b | 0.64 b | 0.67 b | *** | red rose geranium [FL] |
| Citronellol | 69 | 1207 | 8 | 6.7 ab | 5.5 a | 5.4 a | 11.3 b | 5.3 a | 7.8 ab | 11.8 b | 3.1 a | * | floral citrus [FL] |
| Geraniol | 69 | 1235 | 4 | 2.1 ab | 1.6 a | 7.4 d | 8.2 e | 7.5 de | 2.4 b | 4.8 c | 1.7 ab | *** | sweet floral citrus [FL] |
| Citronellol Acetate [x] | 43 | 1333 | | 3.1 d | 2.0 bcd | 2.4 cd | 0.7 ab | 1.6 abc | 0.5 a | 1.7 abc | 1.4 abc | ** | |
| Hydrocinnamyl Acetate [x] | 118 | 1337 | | 1.2 bc | 1.1 bc | 1.4 c | 1.1 bc | 1.4 c | 0.8 b | 0.0 a | 2.2 d | *** | |
| β-Damascenone | 69 | 1376 | 0.002 | 7.1 a | 19.1 d | 15.5 c | 13.7 bc | 11.4 b | 8.4 a | 12.7 bc | 8.4 a | *** | sweet fruity plum [FR] |
| Verdyl Acetate [x] | 66 | 1407 | | 7.4 | 3.8 | 4.7 | 7.4 | 3.7 | 6.5 | 4.1 | 3.0 | ns | |
| Caryophyllene | 93 | 1437 | 64 | 1.9 a | 2.5 a | 1.4 a | 1.9 a | 2.6 ab | 1.6 a | 4.1 b | 1.6 a | * | |
| β-Farnesene | 41 | 1458 | | 2.8 a | 2.6 a | 5.9 b | 2.7 a | 2.8 a | 2.0 a | 3.4 a | 2.8 a | * | |
| β-Ionone | 177 | 1473 | 7 | 0.0 a | 0.2 b | 1.4 d | 0.3 c | 0.2 bc | 0.0 a | 0.0 a | 0.0 a | *** | woody powdery floral [W] |
| α-Farnesene | 41 | 1503 | 87 | 1.0 a | 1.0 a | 2.6 b | 0.6 a | 0.9 a | 0.8 a | 0.8 a | 1.0 a | *** | |
| Nerolidol | 69 | 1562 | 10 | 5.8 d | 6.4 d | 2.7 bc | 10.4 e | 3.8 c | 3.5 c | 1.4 a | 2.1 ab | *** | woody floral citrus [W] |
| 2,3-Dihydrofarnesol [x] | 69 | 1696 | | 5.2 abc | 10.9 cde | 9.0 cde | 14.4 e | 6.4 bcd | 11.9 de | 1.0 ab | 0.2 a | *** | |
| Farnesol [x] | 69 | 1715 | 60000 | 2.9 a | 2.4 a | 11.0 b | 2.9 a | 1.0 a | 3.2 a | 1.6 a | 2.5 a | *** | |
| | | | | | | | Carbonyl compounds | | | | | | |
| Acetaldehyde | 29 | 538 | 5000 | 709 a | 1292 b | 2428 c | 1051 b | 1139 b | 643 a | 1126 b | 3862 d | *** | |
| 3-Methylbutanal | 44 | 634 | 0.2 | 0.7 | 0.4 | 0.7 | 0.6 | 0.6 | 0.8 | 0.6 | 1.0 | ns | |
| 2-Methylbutanal | 41 | 650 | 12.5 | 0.4 | 1.0 | 0.4 | 0.4 | 0.4 | 0.4 | 0.4 | 1.4 | ns | |
| Benzeneacetaldehyde | 91 | 1005 | 4 | 4.8 bc | 0.8 a | 1.0 a | 4.6 b | 4.6 b | 4.3 b | 5.3 c | 1.0 a | *** | green floral honey [FL] |
| Acetophenone [x] | 105 | 1040 | 65 | 0.0 a | 0.0 a | 0.6 c | 0.9 d | 0.0 a | 0.0 a | 0.0 a | 0.2 b | *** | sweet pungent chemical [C] |
| Nonanal | 57 | 1083 | 1 | 3.5 | 2.8 | 4.5 | 3.7 | 3.6 | 3.9 | 4.0 | 4.1 | ns | sweet floral [FL] |
| Decanal | 43 | 1183 | 0.1 | 2.8 abc | 4.6 bc | 2.0 ab | 2.4 ab | 2.3 ab | 2.1 ab | 1.6 a | 5.2 c | * | aldehydic citrus floral [FL] |
| Dodecanal | 57 | 1392 | 2 | 4.0 | 4.7 | 3.5 | 3.2 | 4.2 | 2.5 | 3.3 | 2.8 | ns | citrus green floral [FL] |

**Table 5.** *Cont.*

| [µg/L] | m/z | LRI [2] | Threshold [3] | *S. cerevisiae* Safbrew T-58 | *S. paradoxus* CBS7302 | *S. kudriavzevii* CBS3774 | *Wickerhammomyces anomalus* CBS5759 | *Hanseniaspora uvarum* CBS2768 | *Dekkera bruxellensis* CBS3429 | *Zygosaccharomyces bailii* CBS729 | *Torulaspora delbrueckii* D10 | Sig. [1] | GC-O descriptors [4] |
|---|---|---|---|---|---|---|---|---|---|---|---|---|---|
| | | | | | | | Sulphur compounds | | | | | | |
| Dimethyl disulfide | 94 | 734 | 0.16 | 0.0 a | 0.6 bc | 0.3 abc | 0.9 c | 1.9 d | 0.2 ab | 0.5 abc | 0.5 abc | *** | sulfurous vegetable onion [V] |
| Methional | 48 | 866 | 0.2 | 0.6 a | 0.9 ab | 1.8 def | 1.6 cde | 2.0 ef | 1.2 bc | 2.2 f | 1.4 bcd | *** | boiled potatoes [V] |
| Dimethyl trisulfide | 126 | 933 | 0.005 | 0.14 a | 0.14 a | 0.17 a | 0.34 b | 0.22 a | 0.17 a | 0.15 a | 0.16 a | ** | sulfurous cooked onion [V] |
| Benzothiazole | 135 | 1196 | 80 | 1.44 ab | 1.15 a | 2.03 ab | 4.66 c | 1.09 a | 7.30 d | 3.59 bc | 1.93 ab | *** | |
| Furan compounds | | | | | | | | | | | | | |
| 2-Furanmethanol | 98 | 832 | 2000 | 0.6 cd | 0.4 bcd | 0.0 a | 0.7 de | 0.3 bc | 0.1 ab | 1.0 e | 0.4 bc | *** | |
| | | | | | | | Carboxylic acids | | | | | | |
| Hexanoic acid | 60 | 967 | 3000 | 16.9 b | 13.6 b | 31.3 | 14.6 b | 16.0 b | 16.8 b | 4.5 a | 1.5 a | *** | |
| Octanoic acid | 60 | 1157 | 3000 | 191 c | 127 b | 87 b | 126 b | 128 b | 186 c | 16 a | 98 b | *** | |
| n-Decanoic acid | 60 | 1341 | 10000 | 1471 cd | 836 bc | 1635 d | 864 bc | 451 ab | 1382 cd | 114 a | 630 ab | *** | |
| Dodecanoic acid | 60 | 1545 | 10000 | 11 abc | 7 ab | 17 bc | 13 abc | 6 ab | 19 bc | 2 a | 23 c | * | |
| | | | | | | | Lactones | | | | | | |
| γ-Nonalactone [x] | 85 | 1326 | 30 | 9.8 ab | 15.2 b | 11.1 ab | 11.8 ab | 10.5 ab | 10.8 ab | 22.6 c | 9.1 a | *** | |
| γ-Decalactone | 85 | 1443 | 11 | 8.3 bc | 4.6 a | 9.4 c | 6.7 b | 4.1 a | 3.9 a | 4.5 a | 7.1 b | *** | fresh fruity sweet [FR] |

[1] Significance; *, **, *** indicates significance at a level of 0.05–0.01, 0.01–0.005 and <0.005, respectively, by the least significant difference, ns—not significant. Values with different superscript roman letters (a–f) in the same row indicate statistical differences, according to the Duncan test ($p < 0.05$). [2] LRI—linear retention index; the amount of components was determined. [3] Threshold in beer [31]. OAV > 1. [4] GC-O descriptors perceived at the sniffing port of the GC-O. [x]—determined semi-quantitatively by measuring the relative peak area of each identified compound, according to the NIST database, in relation to that of the chemically similar standard. Aroma group of detected aroma descriptors was signed by letters in brackets—fruity (FR), floral (FL), roasted (R), herbaceous (H), woody (W), vegetal (V), earthy (E), animal (A) and chemical (C). SD < 5%.

The roasted notes dominant in the unfermented wort were significantly reduced. Yeast activity plays an important role on the levels of Strecker aldehydes. In fact, they are reduced to alcohols during fermentation, and on the other hand, the $SO_2$ formed during the process can bind these molecules, thus contributing to a reduction in the perceived "aged character" [33]. In beers, two higher alcohols—2-methylpropanol and 3-methylbutanol—were responsible for roasted notes [Table 3]. Their aroma was described as bready, in higher concentrations of alcoholic or solvent. Higher alcohols are, after ethanol, the volatile compounds found in beers in the largest amounts. They are formed during fermentation from the corresponding amino acids (except 1-propanol). Their influence on the aroma of beer is not only direct, but they are also important precursors for other aroma components, e.g., esters [3]. The higher concentrations of the compounds discussed were found in beers obtained with the participation of *W. anomalus*, the least after fermentation with *Z. bailii*. The higher alcohol, the odour of which was one of the most intense during olfactometric analyses, was 2-phenylethanol [Table 3]. In beers obtained with non-*Saccharomyces* yeast—*W. anomalus*, *Ha. uvarum* and *D. bruxellensis*, the intensity of the phenylethanol aroma exceeded 2.5 points, making the rose notes of this compound the most intense aroma in the samples. Similar results were obtained by Lehnhardt et al. [34] during olfactometric analyzes of lager beers; therefore, phenylethanol should be considered one of the most important volatile components affecting the aroma of beers. Our research has additionally shown that the use of non-*Saccharomyces* cultures in the production of beers can increase the impact of this component on the aroma.

Esters were the most numerous group of volatile compounds in the analyzed beers, but they also had the strongest influence on the aroma. Of the 23 esters detected, 16 were odour-active compounds (detected by GC-O), and most of them were ethyl esters [Table 3]. Although volatile esters are only trace compounds in fermented beverages such as beer, they are extremely important for the aroma profile. The most important aroma-active esters in beer are ethyl acetate (apple-like aroma), isoamyl acetate (fruity, banana aroma), ethyl hexanoate and ethyl octanoate (sour apple), and phenyl ethyl acetate (flowery, roses, honey) [35]. Esters occurring in the largest amounts in the samples did not always significantly affect their aroma due to higher aroma thresholds. The best example of this kind of phenomenon was ethyl acetate, the ester formed during ethanol fermentation in the largest amounts [3]. Thresholds of 5 mg/L were exceeded in five beer samples; however, olfactometric analysis enabled the detection of this component only in three samples, with the average intensity not exceeding one unit. This could be related to the co-elution of this ester with a retention time similar to that of the solvent—ethanol. This can be confirmed by the detection of ethyl acetate aroma in beer obtained with the use of *Z. bailii*, containing smaller amounts of the ester compared to other samples, but at the same time smaller amounts of ethanol.

The obtained beers differed significantly in terms of the most intensively detected esters, which could significantly affect their sensory characteristics (Table 4). The ester with the highest intensity (top three) in the tested samples was ethyl hexanoate, with a pleasant aroma of red apple. Ethyl dihydrocinnamate with a rum and plum aroma, as well as ethyl 4-methylpentanoate and ethyl 2-methylbutyrate with the aroma of apple and red fruit, were also characterized by a very high intensity of aroma. The significant influence of the above esters on the aroma of beer has already been demonstrated by Lehnhardt et al. [34]. The use of yeast culture significantly modified the composition of esters in the obtained beers. The yeast *Saccharomyces* is not as a significant producer of esters as non-*Saccharomyces* yeast [4], and this is especially visible in the case of beers obtained with the participation of *S. cerevisiae* and *S. paradoxus*. A characteristic feature of the beers obtained with the use of *S. cerevisiae* Safbrew was a high concentration of ethyl 2-methylbutanoate and ethyl pentanoate, and because these compounds are characterized by a relatively low threshold, they belonged to the compounds with the strongest influence on the aroma of these samples (respectively, 1st and 3rd compound with the highest aroma intensity). Similar to ethyl hexanoate (2nd most intense compound in these samples), they had a red apple aroma. It

was also one of the two analyzed beers, where the esters were characterized by a greater intensity of aroma than phenylethanol. The second was beer produced with *T. delbrueckii* yeast, where ethyl hexanoate and ethyl dihydrocinnamate were two main odour-active components. At the same time, it was the beer in which the most different esters were detected during GC-O analyses, 12 out of 16 [Tables 3 and 4]. *T. delbrueckii* strains are considered good producers of esters, which is why they are proposed as cultures that can enrich the aroma of beer with fruity and floral notes [36]. The highest total ester intensity was found in beers obtained with the participation of *W. anomalus* and *Ha. uvarum* yeasts. The most fruit and flower aromas were also found in both beers. Among the different cultures of *Saccharomyces*, the *S. kudriavzevii* yeast introduced the most aromatic–active esters into the beer. This confirms our previous research, in which yeast from this species had a positive effect on the sensory characteristics of semi-sweet white wines, among others due to the relatively large amount of esters produced [37]. In low-alcohol beers produced with the use of *Z. bailii* yeast, a slightly lower intensity of esters was found than in other beers. The profile of key aroma esters was also different. During the olfactometric analysis, only nine esters were detected, among which, ethyl 2-methylbutyrate, ethyl 4-methylpentanoate and ethyl dihydrocinnamate were the most intense.

Esters as well as terpenes introduce fruity and floral aromas to beer [38]. HSME-GC-MS analysis showed the presence of 16 terpenes and their derivatives in the samples, six of which were odour-active (Tables 4 and 5). Four of them are characterized by a fruity-floral aroma—cis-rose oxide (rose petals aroma), citronellol (citrus aroma), geraniol (floral-citrus aroma) and damascenone (plum aroma), other two—β-ionone and nerolidol by herbal-woody scents. Methyl geranate (aroma-active component of wort) and humulene present in unfermented hopped wort were not detected in beer samples, and several others showed significant reductions after fermentation (e.g., α-farnesene concentration decreased below the aroma threshold). King and Dickinson [38] showed that brewer's yeast can convert some terpenes to others. For example, the monoterpene alcohols geraniol is converted to citronellol, and humulene is converted to caryophyllene. The lager yeast also produced acetate esters of geraniol and citronellol. Beers obtained with the participation of yeasts *W. anomalus* CBS5759 and *S. kudriavzevii* CBS3774 were characterized by the highest amount of terpenes, as well as the highest aroma intensity of these compounds (Tables 4 and 5). This may mean that they show not only the greatest ability to bioconvert these compounds but also β-glucosidase activity. This enzyme converts terpenes from bound to free form (in which they are found, for example, in hops), thanks to which they begin to affect the aroma [39]. β-Damascenone was identified as the compound with the highest odour activity in wort and beer. It was also one of the terpenes whose content increased after fermentation (over three times) as well as aroma intensity—from 0.6 in the wort to even 1.8 in beers obtained with *W. anomalus* and *D. bruxellensis*. This compound is characterized by a very low odour threshold in water (20–90 ng/L) and can be found in naturally aged beer in a concentration up to 210 µg/L. β-Damascenone can be produced by some non-*Saccharomyces* yeasts during the fermentation processes of wine and beer, such as *Cyberlindnera saturnus*, *Debaryomyces hansenii*, *Hanseniaspora uvarum*, *Metschnikowia pulcherrima*, and *Wickerhamomyces anomalus* [40]. The second most important aroma-active terpene in the analyzed beers was cis-rose oxide. It was present in a rather low concentration (0.5–0.7 µg/L), but due to the low odour threshold, it was detected in almost all samples (the highest intensity of *D. bruxellensis* beers—1.6). Fermentation studies with a model that contained deuterated water revealed that yeast is capable of reducing the precursor 3,7-dimethyl octa-2,5-dien-1,7-diol (geranyl diol I) yielding 3,7-dimethyl-5-octen-1,7-diol (citronellyl diol I) that gives rise to cis- and trans-rose oxide after acid-catalyzed cyclization. The presence of cis-rose oxides in fermented beverages can therefore be attributed to the reductive yeast metabolism during fermentation [41]. It is generally believed that during fermentation, the geraniol from the hops is converted by the yeast into citronellol [39]. In the case of beers obtained with the participation of *S. cerevisiae*, *S. paradoxus*, *D. bruxellensis* and *T. delbrueckii* yeasts, such a bioconversion

occurred (Tables 4 and 5). However, in the remaining four, the level of geraniol remained unchanged and even increased, thus remaining above the aroma threshold. The increase in the level of citronellol was therefore not associated with a reduction in the amount of this compound. According to Takoi et al. [39], this increase in β-citronellol might be partly explained by an occurrence of glycosidically bound flavor precursor and a glucoside hydrolase activity secreted from yeast. Nerolidol was also a terpene that appeared after fermentation, becoming the odour-active component in three beers. Nerolidol accumulation could be stimulated by the inhibition of sterol biosynthesis. Under anaerobic conditions, nerolidol formation would be stimulated, whereas under microaerobic conditions the sterol biosynthesis pathway mostly occurs [42].

The floral-fruity notes in beer were also associated with the presence of aldehydes such as nonanal, decanal and dodecanal. These compounds have a low aroma threshold and are formed as a result of lipid oxidation, and their presence in the amount of several μg/L of beer is its natural feature [43]. All analyzed beers contained the compounds in question in amounts exceeding the aroma threshold. The highest odour intensity of these components was found in samples obtained with *Saccharomyces*, *T. delbrueckii* and *D. bruxellensis* yeasts. Malt and hops contribute to the lipid and fatty acid precursors that can then potentially be transformed by yeast also into other important groups of aroma components of beer— lactones [44]. Most commonly reported lactones in alcoholic beverages are γ and δ-lactones (five and six membered rings, respectively) with their potent oily, stone fruit, and coconut aroma qualities. We detected two different γ-lactones in the samples. As in the case of aldehydes, the sweet, fruity aroma of this γ-decalactone was found during olfactometric analysis in beers fermented with *Saccharomyces* and *T. delbrueckii* yeasts.

Currently, in the brewing industry, more and more attention is paid to the determination of volatile sulfur compounds due to their low aroma thresholds. However, there is still very little research on this subject, and the available articles date back many years. These compounds, present in beer in higher concentrations, can contribute to the formation of unpleasant flavors and can be used as indicators during fermentation [45]. Three sulfur compounds (methional, dimethyl disulfide (DMDS) and dimethyl trisulfide (DMTS)) were found in the analyzed beers, all significantly affecting the aroma of the beverages. As it was already described earlier, all of them were present in the wort. After fermentation, their level decreased but remained above the threshold. The sample in which all these three compounds were detected during olfactometric analyses was beer obtained with the use of *D. bruxellensis* yeast [Tables 4 and 5]. In low-alcohol beer produced by *Z. bailii* yeast, methional was the main odour-active component [Tables 4 and 5]. As it should be assumed, the ability to reduce this compound during fermentation depends on the available carbon sources. The presence of the discussed sulfur compounds is associated with the microbial decomposition of sulfur amino acids—cysteine and methionine [46]. DMTS is characterized by a high potential for introducing sulfuric notes into beers and it is a component of hops [47]. DMTS is recognized, e.g., through aromas such as onion, boiled cabbage, and garlic. These characteristics also resemble DMS (dimethyl sulfide), but they are more intense [48]. This compound is produced during the boiling of wort with hops, and its precursor is S-methylcysteine sulfoxide [49]. Gij et al. [50] in their research proposed two additional sources of DMTS in aged beers: 3-methylthiopropanaldehyde and its reduced form, 3-methylthiopropanol. There is also a positive effect of the presence of sulfides, mainly on the level of trans-nonenal (a wet paper/cardboard note) in beers. They work mainly by masking the aldehyde aroma in the final product [51].

The two main odour-active volatile phenols in beer are 4-vinylguaiacol (4-vinyl-2-methoxyphenol) (4VG) and 4-vinylphenol (4VP) [52]. Despite historically being cataloged as a phenolic off-flavor (POF) (24) in bottom-fermented beers, 4VG is a well-known contributor to the characteristic aroma in top-fermented beers, including Belgian white (brewed with unmalted wheat), German Weizen, and rauch (brewed with malted wheat) [53]. The analyzed beers were characterized by a wide range of 4VG amounts from 3.5 (*Ha. uvarum*) to 109.8 μg/L (*D. bruxellensis*). Due to the low aroma threshold (3 μg/L), the OAV of one

was exceeded in all beers. However, the olfactometric analysis showed a clove-like aroma in four samples—*S. cerevisiae*, *T. delbrueckii*, *W. anomalus*, and *D. bruxellensis* with the highest intensity (1.2) [Table 4]. Members of the genus *Brettanomyces/Dekkera* are yeasts known to produce large amounts of volatile phenols. On the one hand, they are considered dangerous spoilage of fermented beverages such as wine. On the other hand, this feature is used to obtain the typical aroma of Belgian beers—lambic and gueuze [54].

*3.4. Sensory Evaluation of Beers*

The last stage of the research was related to the sensory analysis of the obtained beers. The characteristics of the aroma were studied using the QDA method. On a 9-point scale, the intensity of aromas was determined and grouped into nine categories (fruity, floral, roasted, herbal, woody, vegetable, earthy, animal and chemical), the same as those used to interpret the results of the olfactometric analysis.

Compared to hopped unfermented wort, the intensity of fruit and floral aromas (associated with fermentation products) increased significantly, and the intensity of roasted aromas decreased [Figure 4]. Beers obtained with the use of yeasts *W. anomalus* and *S. kudriavzevii* were characterized by the highest intensity of fruit and floral aromas. Among the detected notes, notes of red apple and strawberries dominated. The greatest contributors to strawberry fruit aroma are ethyl butanoate, ethyl hexanoate, methyl butanoate and methyl hexanoate among esters; 2,5-dimethyl-4-hydroxy-3(2H)-furanone (DMHF) and 4-methoxy-2,5-dimethyl-3(2H)-furanone (DMMF) among furanones; linalool and nerolidol among terpenes; and methanethiol among sulfur compounds [55]. The first two compounds were characterized by a high intensity of aroma in the discussed samples (Table 3). Low-alcohol beer was characterized by the lowest intensity of the discussed notes, which confirms the large impact of the fermentation process on this type of aroma. Roasted notes remained relatively intense in beers obtained with *Ha. uvarum* and *S. kudriavzevii* yeast. The aroma of beer obtained with the participation of *Ha. uvarum* yeast was characterized as porter-like, with notes of honey.

Herbal and woody notes with the highest intensity were detected in beer fermented with *D. bruxellensis* yeast [Figure 4]. It should be assumed that this was due to the presence of a large amount of volatile phenols, such as 2-methoxy-4-vinylphenol, which are characterized by aromas of cloves and spices [53]. To a lesser extent, these aromas were influenced by the hop components, because, as already stated, relatively low hopping of the wort was used. In addition, during the olfactometric analyses, the hop components introduced mainly fruit (plum) and floral (rose, geranium) notes. Vegetable notes were characteristic of low-alcohol beers obtained with the use of *Z. bailii* yeast. As already stated, they were mainly associated with a large amount of methional. In addition, a small amount of other components typical of fermentation exhibited such strong aromas. The resulting beers were characterized by a low intensity of animal, earthy and chemical notes. All of them can impair the sensory characteristics of beer.

We observed differences between the results of olfactometric and sensory analysis, which may be related to the creation of new aromas by overlapping aroma notes. Aroma compounds with similar attributes often have additive interactions that lower the thresholds of the individual compounds. Moreover, in certain cases, aroma compounds exert stronger than anticipated effects by interacting synergistically or antagonistically (masking effect) [56].

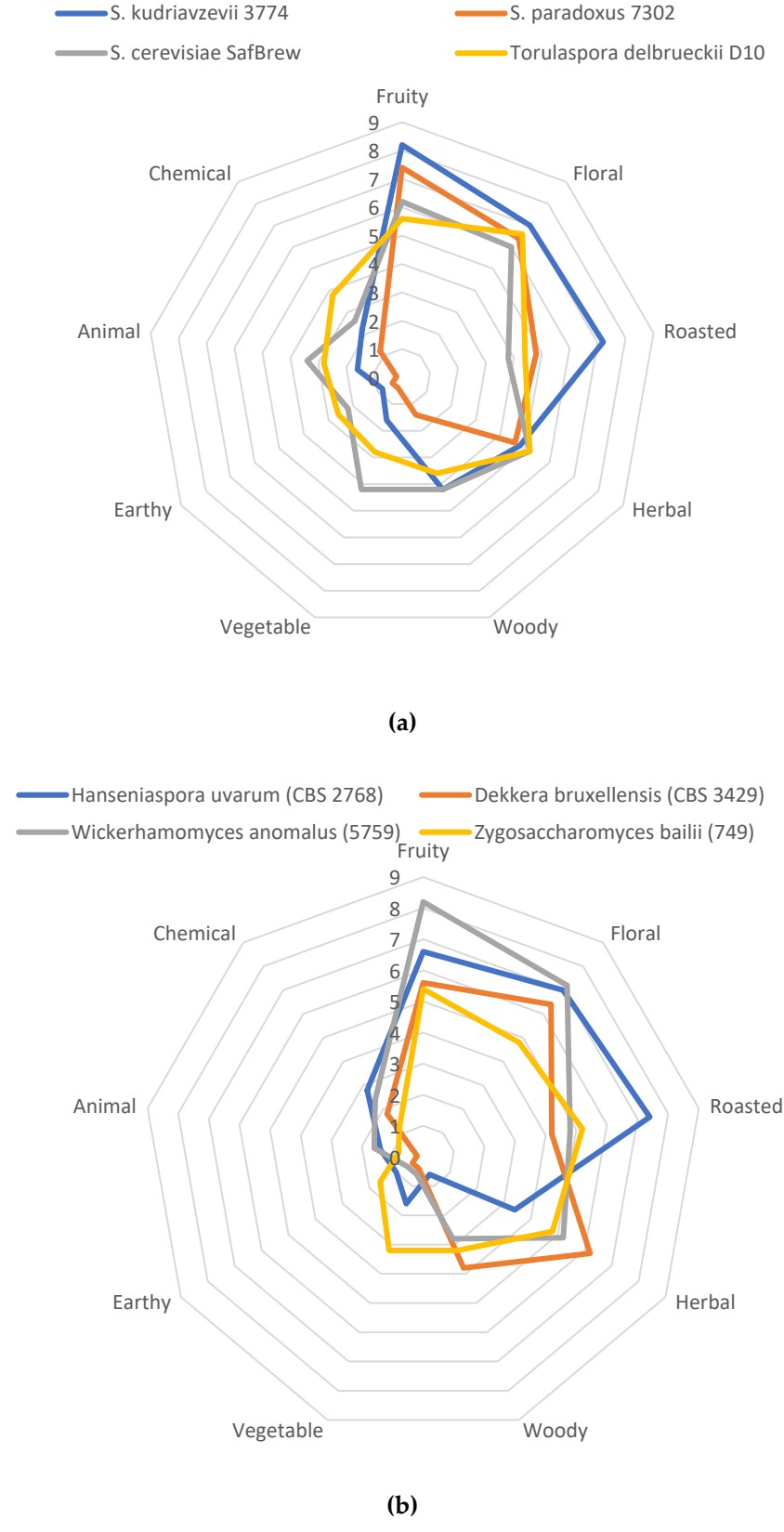

**Figure 4.** Sensory analysis (QDA) of produced beers with non-conventional yeasts. (**a**)—*Saccharomyces* yeast and *Torulaspora delbrueckii*; (**b**)—other non-*Saccharomyces* yeast.

## 4. Conclusions

Our research for the first time dealt with the subject of odour-active components in beers obtained with the use of various unconventional yeasts, both other representatives of the genus *Saccharomyces* and non-*Saccharomyces*, often used in the experiments of other researchers. We showed large differences between the key aroma components depending on the culture of microorganisms used. We detected 40 different compounds that have an active impact on the creation of the aroma of beers, among which the most important are β-phenylethanol, ethyl hexanoate, ethyl 4-methylpentanoate, ethyl dihydrocinnamate and β-damascenone. We also found the presence of components specific to the yeast strain used, such as 2-methoxy-4-vinylphenol, γ-decalactone, methional, nerolidol and others. Among the analyzed yeasts, *S. kudriavzevii* and *W. anomalus* should be distinguished, which produced beers with intense fruity and floral aromas and were also characterized by favorable features for brewing. The *Z. bailii* strain also turned out to be interesting as a potential starter culture for the production of low-alcohol beers, significantly differing in sensory characteristics from the standard ones. In the coming years, further research will be undertaken to present and examine the possibilities of using different activities of these cultures in the production of beer and other fermented beverages.

**Author Contributions:** Conceptualization, P.S.; methodology, A.P. and P.S.; software, A.P. and P.S.; validation, A.P. and P.S.; formal analysis, A.P. and P.S.; investigation, P.S. and A.P.; resources, P.S. and A.P.; data curation, P.S. and A.P.; writing—original draft preparation, P.S. and A.P.; writing—review and editing, P.S. and A.P.; visualization, P.S. and A.P.; supervision, P.S; project administration, P.S.; funding acquisition, P.S. All authors have read and agreed to the published version of the manuscript.

**Funding:** This research was financed by The Ministry of Science and Higher Education of Poland, as a part of the Science Subsidy number 070013-D020.

**Institutional Review Board Statement:** Not applicable.

**Informed Consent Statement:** Not applicable.

**Data Availability Statement:** Data on the compounds are available from the authors.

**Conflicts of Interest:** The authors declare no conflict of interest.

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
