# Peer review of "The Influence of Different Non-Conventional Yeasts on the Odour-Active Compounds of Produced Beers"

_applsci, doi:10.3390/app13052872_

Round 1

Reviewer 1 Report

Overall, the manuscript is clearly written and describes a thorough and interesting piece of work. Some improvements could be made however. More discussion of potential interactions could improve interpretation of the results, particularly in section 3.2, in later sections this is adequately addressed. Some minor grammatical errors are also noted, though these do not generally detract from the message.

Some specific edits and suggestions are as follows.

Methods:

Section 2.5 should specify the apparatus were HPLC for clarity.

Section 2.6 and 2.7 utilise different SPME fiber phases for GC-FID and GC-MS vs GC-O. The selectivity and loading capacities of 100 um PDMS vs DVB/CAR/PDMS are quite different, therefore how are you able to compare the intensity of the aromas detected by GC-O with the quantitative/semi-quantitative chemistry data?

Results and Discussion:

Table 1 – roman letters have not been made superscript as the table footer says.

Tables 2, 3 and 4 – please add an explanation of the letters in square brackets to the footers.

Please check the accuracy of lines 403-405. Saccharomyces are significant producers of esters to this reviewers knowledge, and nowhere in the cited paper does Gschaedler imply this.

It is unclear what the authors are referring to by “the first culture” in line 405.

Figures 4 and 5 – while this reviewer appreciates the complexity of GC-O data, the way these graphs present those data do not make it easy for the reader to interpret. A table or a heat map might be a clearer alternative, and it would be of interest to have all of the information presented in both figures combined into one for better comparisons between Sac and non-Sac yeasts.

Line 438 please rewrite, esters and terpenes result in fruity and floral aromas, not the other way around.

Line 471 the authors endnote library could not find the reference, please put correct reference in.

Line 472 makes the “it is generally believed” statement without providing evidence. Please provide a reference for this statement.

Section 3.4 implies the QDA was more comprehensive than the 9 aroma categories presented. It would be of great interest to understand the nuanced differences within those categories as well as the main differences due to yeast strain.
Additionally, combining the results of the Sac and non-Sac yeasts into a single graph would be easier for the reader to interpret. It is also noted that there are no references to figure 6 in the text, these should be added.

References:

Several references have first letters in the titles capitalised, while the majority do not. Please edit for consistency.

Author Response

Reviewer 1

Authors would like to express their sincere thankfulness to the editor and reviewers for their sincere evaluation of our paper and their stimulating comments. All their remarks and advices are followed. In the returned, revised manuscript, all the introduced changes are marked.

Below are our answers to individual comments.

Overall, the manuscript is clearly written and describes a thorough and interesting piece of work. Some improvements could be made however. More discussion of potential interactions could improve interpretation of the results, particularly in section 3.2, in later sections this is adequately addressed. Some minor grammatical errors are also noted, though these do not generally detract from the message.

Some specific edits and suggestions are as follows.

Methods:

Section 2.5 should specify the apparatus were HPLC for clarity.

It was already discribed in the text - Shimadzu (Kyoto, Japan) NEXERA XR apparatus with an RF-20A refractometric detector was used for sugars detection and Perkin-Elmer (USA) FLEXAR chromatograph with a UV-Vis detector was used for organic acids detection.

Section 2.6 and 2.7 utilise different SPME fiber phases for GC-FID and GC-MS vs GC-O. The selectivity and loading capacities of 100 um PDMS vs DVB/CAR/PDMS are quite different, therefore how are you able to compare the intensity of the aromas detected by GC-O with the quantitative/semi-quantitative chemistry data?

We used the phases that worked best for the method. For the olfactometric analysis, a DVB/CAR/PDMS fiber was used, 2 cm long (the only one of this length by Supelco). Numerous earlier publications confirmed its beneficial properties when it comes to the GC-O method (Rega B, Fournier N, Guichard E. Solid phase microextraction (SPME) of orange juice flavor: odor representativeness by direct gas chromatography olfactometry (D-GC-O). J Agric Food Chem. 2003 Nov 19;51(24):7092-9. doi: 10.1021/jf034384z. PMID: 14611177.). Our earlier tests showed also that the obtained results using HS-SPME are very similar to that obtained using SAFE analysis. Thus HS-SPME-GC-O can be used in the olfactometric analysis of beer instead of the SAFE method. The benefits of this are very large - less sample to be analyzed, faster sample preparation, the ability to perform more repetitions, etc. (Witrick, K.; Pitts, E.R.; O’Keefe, S.F. Analysis of Lambic Beer Volatiles during Aging Using Gas Chromatography–Mass Spectrometry (GCMS) and Gas Chromatography–Olfactometry (GCO). Beverages 2020, 6, 31. https://doi.org/10.3390/beverages6020031). However, our previous research, as well as that of other scientists analyzing volatile compounds in fermented beverages, 1 cm long PDMS fiber is much better suited for quantitative analysis. It showed adequate linearity, sensitivity, repeatability and intermediate precision. It can also minimize the displacement effect caused by ethanol in the quantitative determination of volatile profile of beers and other fermented beverages (Hernandes, K. C., Souza-Silva, É. A., Assumpção, C. F., Zini, C. A., & Welke, J. E. (2019). Matrix-compatible solid phase microextraction coating improves quantitative analysis of volatile profile throughout brewing stages. Food Research International. doi:10.1016/j.foodres.2019.04.048)

Results and Discussion:

Table 1 – roman letters have not been made superscript as the table footer says.

Corrected

Tables 2, 3 and 4 – please add an explanation of the letters in square brackets to the footers.

Corrected

Please check the accuracy of lines 403-405. Saccharomyces are significant producers of esters to this reviewers knowledge, and nowhere in the cited paper does Gschaedler imply this.

Corrected

It is unclear what the authors are referring to by “the first culture” in line 405.

Corrected

Figures 4 and 5 – while this reviewer appreciates the complexity of GC-O data, the way these graphs present those data do not make it easy for the reader to interpret. A table or a heat map might be a clearer alternative, and it would be of interest to have all of the information presented in both figures combined into one for better comparisons between Sac and non-Sac yeasts.

We replaced the Figures 4 and 5 by the table with heat-map. I hope that now the results are easier to interpret.

Line 438 please rewrite, esters and terpenes result in fruity and floral aromas, not the other way around.

Corrected

Line 471 the authors endnote library could not find the reference, please put correct reference in.

Corrected

Line 472 makes the “it is generally believed” statement without providing evidence. Please provide a reference for this statement.

Corrected

Section 3.4 implies the QDA was more comprehensive than the 9 aroma categories presented. It would be of great interest to understand the nuanced differences within those categories as well as the main differences due to yeast strain.

Each of the evaluators, in addition to awarding points in each of the aroma descriptors, also recorded the general characteristics of the beer's aroma. They have been inserted/modified in the text.

Additionally, combining the results of the Sac and non-Sac yeasts into a single graph would be easier for the reader to interpret.

Unfortunately, the combination of these two figures significantly affects their readability and makes it difficult to interpret them. We will try to put both of them on one page in the final version of the article to facilitate the comparison of the QDA results of beers obtained with Saccharomyces and non-Saccharomyces yeasts

It is also noted that there are no references to figure 6 in the text, these should be added.

Corrected

References:

Several references have first letters in the titles capitalised, while the majority do not. Please edit for consistency.

Corrected

Reviewer 2 Report

I was invited for the revision of the manuscript entitled “The influence of different non-conventional yeasts on the 2 odour-active compounds of produced beers”. This paper focuses on the differences on beer aroma due to the use of non-conventional yeasts for the fermentation. The authors did an excellent job providing an accurate experimental design with a comprehensive approach completed by GC, LC, olfactory and sensory analyses. English is fine and I was able to easily read and understand all information. From my point of view the manuscript is eligible to be accepted after minor revision (I have some comments reported below).

Line 30. Both in the abstract and introduction the same sentence was used (The interest in new beer products, which has been growing for several years, has encouraged technologists/brewers to search for innovative raw materials, such as hops, new sources of 10 carbohydrates or yeast). Please change its form in one of the two sections.

Line 69. Nowadays, most of the times the secondary detector is an EI-MS. Please, add it.

Line 122. Please, add details on the calibration (matrix used for the standard dilution, calibration range, linearity, LOD/LOQ).

Line 133. 1g of NaCl in 2 mL of beer means 500g in 1L. The solubility of NaCl is known to be 360g/L so this means that non all salt will be dissolved. Please check if it was what you did or a typo.

Line 139. 40 minutes at 40°C I am confident that the fiber will be saturated and it will not give you a representative snapshot of the beer aroma. Did you evaluate it? Those are also abundant analytes...

Line 170. Same of previous comment on salt concentration.

Line 281. These carbonyls are mostly related to Millard reaction but not only to this process. For instance, they can also come from raw materials or other routes.

Table 2. Many well-known aroma descriptors are missing. Benzaldehyde is almond-like, hexanoic acid is goat-like and so on.. I invite the authors to do a more careful literature exploration.

Line 350. Guedes de Pinho? Typo?

Figures 4 and 5. Unfortunately, these aroma intensities can be affected by the magnification due to the presence of other compounds. For instance, Ferreira evaluated this effect in this paper concerning wine DOI: 10.1016/B978-0-08-102067-8.00008-7. Can you demonstrate that your values are not affected by this kind of phenomena?

Author Response

Reviewer 2

Authors would like to express their sincere thankfulness to the editor and reviewers for their sincere evaluation of our paper and their stimulating comments. All their remarks and advices are followed. In the returned, revised manuscript, all the introduced changes are marked.

Below are our answers to individual comments.

I was invited for the revision of the manuscript entitled “The influence of different non-conventional yeasts on the 2 odour-active compounds of produced beers”. This paper focuses on the differences on beer aroma due to the use of non-conventional yeasts for the fermentation. The authors did an excellent job providing an accurate experimental design with a comprehensive approach completed by GC, LC, olfactory and sensory analyses. English is fine and I was able to easily read and understand all information. From my point of view the manuscript is eligible to be accepted after minor revision (I have some comments reported below).

Line 30. Both in the abstract and introduction the same sentence was used (The interest in new beer products, which has been growing for several years, has encouraged technologists/brewers to search for innovative raw materials, such as hops, new sources of 10 carbohydrates or yeast). Please change its form in one of the two sections.

Corrected

Line 69. Nowadays, most of the times the secondary detector is an EI-MS. Please, add it.

Corrected

Line 122. Please, add details on the calibration (matrix used for the standard dilution, calibration range, linearity, LOD/LOQ).

We used a validated method that has been used in analyzes for several publications. For this reason, we will not include these types of parameters in the publication. The appendix presents the most important parameters of the methods.

Concentration range (g L-1)

R2

Slope

LOD (mg L-1)

LOQ (mg L-1)

glucose

0,3125-10

0,999

141542

110,8

335,8

glycerol

0,5-16

0,999

1893,1

39,1

118,6

fructose

0,3125-10

0,999

84512

141

427,3

saccharose

0,3125-10

0,999

111608

176

533,5

maltose

0,3125-10

0,999

91414

17,2

52,2

lactic acid

19,5-625

0,999

1852

1,3

4

acetic acid

39-625

0,999

2460,8

7,5

22,8

succinic

15.6-500

0,999

7737,4

4,8

14,4

Line 133. 1g of NaCl in 2 mL of beer means 500g in 1L. The solubility of NaCl is known to be 360g/L so this means that non all salt will be dissolved. Please check if it was what you did or a typo.

Saturated or even supersaturated solution of sodium chloride is used during Solid Phase Microextraction (SPME), to increase the amount of volatile compounds in the headspace.

Line 139. 40 minutes at 40°C I am confident that the fiber will be saturated and it will not give you a representative snapshot of the beer aroma. Did you evaluate it? Those are also abundant analytes...

40 minutes and 40áµ’C are optimal conditions for SPME extraction for beer as well as other fermented beverages aroma compounds. The method was optimized by us, but of course also other scientists obtained similar results (Martins, C., Brandão, T., Almeida, A., & Rocha, S. M. (2015). Insights on beer volatile profile: Optimization of solid-phase microextraction procedure taking advantage of the comprehensive two-dimensional gas chromatography structured separation. Journal of Separation Science, 38(12), 2140–2148. doi:10.1002/jssc.201401388)

Line 170. Same of previous comment on salt concentration.

Explained above

Line 281. These carbonyls are mostly related to Millard reaction but not only to this process. For instance, they can also come from raw materials or other routes.

But in case of wort most of them is formed during boiling.

Table 2. Many well-known aroma descriptors are missing. Benzaldehyde is almond-like, hexanoic acid is goat-like and so on.. I invite the authors to do a more careful literature exploration.

In table 2 aroma descriptors only for odour-active components perceived at the sniffing port of

the GC-O in our samples were described (superscript 4 in the table).

Line 350. Guedes de Pinho? Typo?

Corrected

Figures 4 and 5. Unfortunately, these aroma intensities can be affected by the magnification due to the presence of other compounds. For instance, Ferreira evaluated this effect in this paper concerning wine DOI: 10.1016/B978-0-08-102067-8.00008-7. Can you demonstrate that your values are not affected by this kind of phenomena?

I have read Ms. Ferreira's chapter on the aroma of wine, but it is more concerned with the influence of various components on the aroma of wine during tasting than with the intensity of individual components during olfactometric analysis. Indeed, when we consume wine/beer, there are numerous interactions between the components (in the mixture) and our senses, creating sensations, flavors, and aromas that are not necessarily characterized by single volatile compounds. In the case of olfactometric analysis, it is much easier, because we first separate the mixture using a chromatograph/column, and only then we detect the pure compounds flowing down to our nose acting as a detector. We also remove all non-volatile components, which are the most abundant in a fermented beverage, and which may bind and limit the volatility of some aromas. In the case of beer, it is also a bit easier than in drinks containing more alcohol, because there are also less volatile compounds, 3-4 or even more times than, for example, in wine. The biggest problem in the discussed method may be insufficient recovery of volatile substances from the analyzed matrix or co-elution (similar retention times) of compounds characterized by high aroma intensity/low threshold. Therefore, it is important to use an appropriate extraction method (SPME works great for beers, and the results can be even better/more sensitive than in the case of the SAFE method), and a separation program so that the retention times of odor-active compounds do not overlap. Preparing for the presented research, we analyzed over 100 samples of commercial beers and those prepared in our laboratories.

Reviewer 3 Report

This study is of scientific interest in the field of brewing industry. The authors studied the application of unconventional Saccharomyces and non-Saccharomyces yeasts into the beer production in order to improve or differentiate the quality of the fermented product, especially its chemical composition and key aroma-active compounds.

The work is original, is appropriately designed, and results are clearly presented in Tables and Figures. However, I consider that the description of results in the text are not properly referenced to the respective figures and tables, thus it is difficult to follow the reading in some parts of the manuscript and the interpretation of results. Apart from that, the work is properly written and presented, although there are minor mistakes in units of certain parameters or expression of results in the Tables or some missing references for abbreviations.

On the other hand, the standard deviations or errors of the results exhibited in Tables and Figures are missing. Additionally, the Duncan`s test (significant differences indicated by different letters) applied to the results presented in Tables should be revised. Please, correct or modify these data accordingly.

Minor revisions:

Line 95: Change "amounts" to "amount" and add the word "of" before "the individual..."

Line 235: Replace "presented studies" by "present study" because the expression is confusing regarding previous citation about S. cerevisiae.

Lines 225-227: Did you mean that strains of the S. kudriavzevii species are present in spontaneous fermentation? I consider that you should write down "S. kudriavzevii species is present in Belgian beers..."

The citation indicates that it is a reference to the species, or did you refer to the strain studied in your work? Please, clarify.

Line 261: Replace the beginning of the sentence with "In the present study".

Line 264: Did you mean "fermentation ability" or "fermentability"? Please, revise this expression.

Lines 293-294: Why did you say "the yeast extract, usually stored long before use". Did you refer to "malt extract" and why did you say that it was stored long time? Please, clarify this hypothesis or suggestion.

Lines 299-300: Please, rewrite this sentence in order to improve the English.

Line 307: Table 1. Please, revise the unit used for ethanol content (g/L) or the values expressed in g/L, which are very low.

Line 307: Table 1. A) Please, correct the spelling of "Saccharose". B) Revise the unit (g/L) or the contents of succinic acid in the beers reported. C) Is the acetic acid measured in g/L or in mg/L? Please, revise the concentrations of acetic acid in the beers reported. D) Statistical significance, according to the Duncan test (p<0.05), should have a relation between the order of the superscript roman letters (a, b, c…) and the decreasing or increasing levels of biomass (and the other parameters analyzed: ethanol, saccharose, fructose, glucose, glycerol, lactic and acetic acids).

Please, revise this analysis to order and facilitate the interpretation. In the case of Maltose, the Duncan's test is ordered, the letters (a, b and c) are assigned to values in a decreasing way; and in Succinic acid, the letters are assigned in an increasing way (of the values).

Please, revise accordingly throughout the manuscript (Tables).

Line 313: Table 2. What do the capital letters between square brackets mean in Aroma descriptors column? Please, add this reference in the footnote of the Table2 or make reference to the section 2.6 in M&M.

Line 313: Table2. The average concentration of aroma components does not have an error or standard deviation (SD). Please, add it to each value. In the same way, add SD to each concentration value in Tables 3 and 4; and to the bars of aroma-active compounds in Figures 2, 4 and 5.

Line 315: Correct the spelling "identified".

Line 321: Add a reference to a figure or table in the discussion of results in this paragraph. Why did you say that "the olfactometric analysis showed the influence of 5 compounds derived from hops on the aroma of the hopped wort – linalool, geraniol, methyl geranate, damascenone and alpha-farnesene." When these compounds exhibited the lowest points. Add a reference to the corresponding Table to values of line 326.

Line 348: Correct by "on the other hand..."

Line 350 and 359: I recommend to cite in the text the Table or figure where these results are shown.

Line 351: Delete the surname of the author after the citation (33).

Line 370: Table 3. Please, change to italics the names of species Zygosaccharomyces bailii and Torulaspora delbrueckii.

Line 372: Change “column” to "row".

Lines 377-378: What did you mean when you say "they were distinguished by various esters, carbonyl compounds, and terpenes"? It is confusing…P lease, revise and rewrite this sentence. And make reference to the Tables where all the compounds are shown.

Lines 380-381: Please, cite the Table or Figure where these results are shown in this work. (Table 3)

Line 415: Add the reference (Figure 4).

Line 426: Please, correct the word "among" with lowercase letter.

Line 436: Correct the "uvarum" with lowercase letter and use italics for species name.

Line 487: Table 4. The OAV subtitle of the table is wrong? Did you want to write down "Threshold"?

Line 489: Change “column” to "row". And change “OAV” to “Threshold” in the subtitle of the respective column.

Line 493: Add a coma after "threshold" or rewrite this sentence.

Lines 520-511: Add in the text the reference to the Table and Figure/s where these results are shown.

Line 538: Add the citation of the Figure where this result (1.2) is shown.

Author Response

Reviewer 3

Authors would like to express their sincere thankfulness to the editor and reviewers for their sincere evaluation of our paper and their stimulating comments. All their remarks and advices are followed. In the returned, revised manuscript, all the introduced changes are marked.

Below are our answers to individual comments.

This study is of scientific interest in the field of brewing industry. The authors studied the application of unconventional Saccharomyces and non-Saccharomyces yeasts into the beer production in order to improve or differentiate the quality of the fermented product, especially its chemical composition and key aroma-active compounds.

The work is original, is appropriately designed, and results are clearly presented in Tables and Figures. However, I consider that the description of results in the text are not properly referenced to the respective figures and tables, thus it is difficult to follow the reading in some parts of the manuscript and the interpretation of results. Apart from that, the work is properly written and presented, although there are minor mistakes in units of certain parameters or expression of results in the Tables or some missing references for abbreviations.

On the other hand, the standard deviations or errors of the results exhibited in Tables and Figures are missing. Additionally, the Duncan`s test (significant differences indicated by different letters) applied to the results presented in Tables should be revised. Please, correct or modify these data accordingly.

Minor revisions:

Line 95: Change "amounts" to "amount" and add the word "of" before "the individual..."

Corrected

Line 235: Replace "presented studies" by "present study" because the expression is confusing regarding previous citation about S. cerevisiae.

Corrected

Lines 225-227: Did you mean that strains of the S. kudriavzevii species are present in spontaneous fermentation? I consider that you should write down "S. kudriavzevii species is present in Belgian beers..."

Corrected

The citation indicates that it is a reference to the species, or did you refer to the strain studied in your work? Please, clarify.

It is refered to species. Corrected

Line 261: Replace the beginning of the sentence with "In the present study".

Corrected

Line 264: Did you mean "fermentation ability" or "fermentability"? Please, revise this expression.

Corrected

Lines 293-294: Why did you say "the yeast extract, usually stored long before use". Did you refer to "malt extract" and why did you say that it was stored long time? Please, clarify this hypothesis or suggestion.

Corrected, of course we meant „malt extract”. Malt extract, due to the high content of sugars, can be stored for a longer period of time and is microbiologically stable. Malt extract is used in smaller breweries, but is not produced there. It is transported from the manufacturer, stored and used as needed.

Lines 299-300: Please, rewrite this sentence in order to improve the English.

Corrected

Line 307: Table 1. Please, revise the unit used for ethanol content (g/L) or the values expressed in g/L, which are very low.

Corrected

Line 307: Table 1. A) Please, correct the spelling of "Saccharose".

Corrected

  1. B) Revise the unit (g/L) or the contents of succinic acid in the beers reported.

Corrected

  1. C) Is the acetic acid measured in g/L or in mg/L? Please, revise the concentrations of acetic acid in the beers reported.

Corrected

  1. D) Statistical significance, according to the Duncan test (p<0.05), should have a relation between the order of the superscript roman letters (a, b, c…) and the decreasing or increasing levels of biomass (and the other parameters analyzed: ethanol, saccharose, fructose, glucose, glycerol, lactic and acetic acids).

Please, revise this analysis to order and facilitate the interpretation. In the case of Maltose, the Duncan's test is ordered, the letters (a, b and c) are assigned to values in a decreasing way; and in Succinic acid, the letters are assigned in an increasing way (of the values).

Please, revise accordingly throughout the manuscript (Tables).

Corrected

Line 313: Table 2. What do the capital letters between square brackets mean in Aroma descriptors column? Please, add this reference in the footnote of the Table2 or make reference to the section 2.6 in M&M.

Corrected

Line 313: Table2. The average concentration of aroma components does not have an error or standard deviation (SD). Please, add it to each value. In the same way, add SD to each concentration value in Tables 3 and 4; and to the bars of aroma-active compounds in Figures 2, 4 and 5.

Due to the very large number of results, we decided not to introduce standard deviations into the tables, which would affect the transparency of the results and make their interpretation difficult for the reader. The SD was below 5%, which is confirmed by the ANOVA results.

Line 315: Correct the spelling "identified".

Corrected

Line 321: Add a reference to a figure or table in the discussion of results in this paragraph. Why did you say that "the olfactometric analysis showed the influence of 5 compounds derived from hops on the aroma of the hopped wort – linalool, geraniol, methyl geranate, damascenone and alpha-farnesene." When these compounds exhibited the lowest points. Add a reference to the corresponding Table to values of line 326.

References were corrected. The sentence was rewritten to better reflect the intention of the Authors

Line 348: Correct by "on the other hand..."

Corrected

Line 350 and 359: I recommend to cite in the text the Table or figure where these results are shown.

Corrected

Line 351: Delete the surname of the author after the citation (33).

Corrected

Line 370: Table 3. Please, change to italics the names of species Zygosaccharomyces bailii and Torulaspora delbrueckii.

Corrected

Line 372: Change “column” to "row".

Corrected

Lines 377-378: What did you mean when you say "they were distinguished by various esters, carbonyl compounds, and terpenes"? It is confusing…P lease, revise and rewrite this sentence. And make reference to the Tables where all the compounds are shown.

Corrected

Lines 380-381: Please, cite the Table or Figure where these results are shown in this work. (Table 3)

Corrected

Line 415: Add the reference (Figure 4).

Corrected

Line 426: Please, correct the word "among" with lowercase letter.

Corrected

Line 436: Correct the "uvarum" with lowercase letter and use italics for species name.

Line 487: Table 4. The OAV subtitle of the table is wrong? Did you want to write down "Threshold"?

Corrected

Line 489: Change “column” to "row". And change “OAV” to “Threshold” in the subtitle of the respective column.

Corrected

Line 493: Add a coma after "threshold" or rewrite this sentence.

Corrected

Lines 520-511: Add in the text the reference to the Table and Figure/s where these results are shown.

Corrected

Line 538: Add the citation of the Figure where this result (1.2) is shown.

Corrected

Reviewer 4 Report

This study mainly investigated the effects of different non-conventional yeasts on beer aroma using chromatographic methods and sensory evaluation. The MS was well written, and the results are meaningful for the high-quality beer production.

My concerns are some details that the authors should take in account:

1. The comparison of extracellular enzyme activity between different yeasts should be considered, such as glycosidase, esterase etc., which plays important role in the production of aroma compounds during fermentation process.

2.  The overall rating for beer sensory evaluation should be supplemented.

3. Line 471, check the cited reference.

Author Response

Reviewer 4

This study mainly investigated the effects of different non-conventional yeasts on beer aroma using chromatographic methods and sensory evaluation. The MS was well written, and the results are meaningful for the high-quality beer production.

My concerns are some details that the authors should take in account:

  1. The comparison of extracellular enzyme activity between different yeasts should be considered, such as glycosidase, esterase etc., which plays important role in the production of aroma compounds during fermentation process.

Enzymatic activity of used strains was discussed in some parts. However we didn’t analyze it during this investigations. However thank You for good idea for future experiments.

  1. The overall rating for beer sensory evaluation should be supplemented.

The overall rating was not a subject of current studies. We checked the influence of the strain on the different aroma parameters. However in my opinion all the analysed beers had good and very unique characteristics. To of them obtained with S. kudriavzevii and W. anomalus we plan to produce on a larger scale in our university brewery.

  1. Line 471, check the cited reference.

Corrected
